# Aberrant regulation of the GSK-3β/NRF2 axis unveils a novel therapy for adrenoleukodystrophy

Pablo Ranea-Robles[1,2] (ID), Nathalie Launay[1,2], Montserrat Ruiz[1,2], Noel Ylagan Calingasan[3], Magali Dumont[4], Alba Naudí[5], Manuel Portero-Otín[5], Reinald Pamplona[5], Isidre Ferrer[6,7,8,9], M Flint Beal[3], Stéphane Fourcade[1,2,*] (ID) & Aurora Pujol[1,2,10,**] (ID)

## Abstract

The nuclear factor erythroid 2-like 2 (NRF2) is the master regulator of endogenous antioxidant responses. Oxidative damage is a shared and early-appearing feature in X-linked adrenoleukodystrophy (X-ALD) patients and the mouse model (*Abcd1* null mouse). This rare neurometabolic disease is caused by the loss of function of the peroxisomal transporter ABCD1, leading to an accumulation of very long-chain fatty acids and the induction of reactive oxygen species of mitochondrial origin. Here, we identify an impaired NRF2 response caused by aberrant activity of GSK-3β. We find that GSK-3β inhibitors can significantly reactivate the blunted NRF2 response in patients' fibroblasts. In the mouse models (*Abcd1⁻* and *Abcd1⁻/Abcd2⁻/⁻* mice), oral administration of dimethyl fumarate (DMF/BG12/Tecfidera), an NRF2 activator in use for multiple sclerosis, normalized (i) mitochondrial depletion, (ii) bioenergetic failure, (iii) oxidative damage, and (iv) inflammation, highlighting an intricate cross-talk governing energetic and redox homeostasis in X-ALD. Importantly, DMF halted axonal degeneration and locomotor disability suggesting that therapies activating NRF2 hold therapeutic potential for X-ALD and other axonopathies with impaired GSK-3β/NRF2 axis.

**Keywords** adrenoleukodystrophy; dimethyl fumarate; GSK-3; NRF2; oxidative stress

**Subject Categories** Genetics, Gene Therapy & Genetic Disease; Metabolism; Neuroscience

## Introduction

Oxidative stress and mitochondrial dysfunction contribute to the onset and progression of age-related neurodegenerative diseases, such as amyotrophic lateral sclerosis, Parkinson's, Huntington's, and Alzheimer's disease (Lin & Beal, 2006). A common theme among these disorders, as well as the prototypic demyelinating disease multiple sclerosis, is axonal degeneration (Li *et al*, 2001; Tallantyre *et al*, 2010).

Endogenous antioxidant responses are controlled by nuclear factor erythroid 2-like 2 (NRF2, encoded by *NFE2L2*), which binds to antioxidant response element (ARE) in the promoter region of target genes, subsequently activating the transcription of genes encoding phase II detoxifying enzymes and cytoprotective defences against oxidative stress (Itoh *et al*, 1997). These genes include heme oxygenase-1 (*HMOX1*), NAD(P)H:quinone oxidoreductase-1 (*NQO1*), and enzymes of glutathione metabolism, such as glutathione S-transferases (*GST*), glutamate-cysteine ligase (*GCL*), and glutathione peroxidases (McMahon *et al*, 2001; Lee *et al*, 2003). NRF2 also regulates proteostasis (Komatsu *et al*, 2010; Pajares *et al*, 2016), neuroinflammation (Innamorato *et al*, 2008; Rojo *et al*, 2010), and bioenergetic homeostasis (Holmstrom *et al*, 2013) in the nervous system, such that activating NRF2-dependent responses initiates a sustained neuroprotective effect in several neurodegenerative disorder models (Kanninen *et al*, 2009; Neymotin *et al*, 2011; Stack *et al*, 2011; Kaidery *et al*, 2013; Lastres-Becker *et al*, 2016). We therefore sought to explore the role of NRF2 pathway in the neurodegenerative processes of X-linked adrenoleukodystrophy (X-ALD; McKusick no. 300100).

This is the most common peroxisomal disease and leukodystrophy with an incidence of 1:15,000 (Kemper *et al*, 2017). It is caused by mutations in the *ABCD1* gene (Mosser *et al*, 1993) located on

1   Neurometabolic Diseases Laboratory, Bellvitge Biomedical Research Institute (IDIBELL), L'Hospitalet de Llobregat, Barcelona, Spain
2   CIBERER U759, Center for Biomedical Research on Rare Diseases, ISCIII, Barcelona, Spain
3   Feil Family Brain and Mind Research Institute, Weill Cornell Medical College, New York, NY, USA
4   UMR S 1127, Inserm, U1127, CNRS, UMR 7225, Institut du Cerveau et de la Moelle épinière, Sorbonne Universités, UPMC Université Paris 06, Paris, France
5   Experimental Medicine Department, University of Lleida-IRB Lleida, Lleida, Spain
6   Department of Pathology and Experimental Therapeutics, Faculty of Medicine, University of Barcelona, L'Hospitalet de Llobregat, Barcelona, Spain
7   Center for Biomedical Research on Neurodegenerative Diseases (CIBERNED), ISCIII, Madrid, Spain
8   Institute of Neurosciences, University of Barcelona, Barcelona, Spain
9   IDIBELL-Bellvitge University Hospital, L'Hospitalet de Llobregat, Spain
10  Catalan Institution of Research and Advanced Studies (ICREA), Barcelona, Spain
    *Corresponding author. Tel: +34 932 60 71 37; Fax: +34 932 60 74 14; E-mail: sfourcade@idibell.cat
    **Corresponding author. Tel: +34 932 60 71 37; Fax: +34 932 60 74 14; E-mail: apujol@idibell.cat

Xq.28, which encodes a peroxisomal transporter that moves very long-chain fatty acids (VCLFA) into the peroxisome for degradation by β-oxidation (van Roermund et al, 2008; Wiesinger et al, 2013). As a consequence, very long-chain fatty acids (VLCFA), especially C26:0, accumulate in tissues and plasma and constitute a pathognomonic biomarker for diagnosis. There are two main forms of the disease (Engelen et al, 2012). First, cerebral adrenoleukodystrophy is present mostly in boys between 5 and 10 years (35–40% of the cases) but also in adolescents and adult men, who present a strong inflammatory demyelinating reaction in central nervous system white matter. Second, adrenomyeloneuropathy occurs in 60% of the cases and affects adult men and heterozygous women over the age of 40 (Engelen et al, 2014). Adrenomyeloneuropathy is characterized by peripheral neuropathy and distal axonopathy involving corticospinal tracts of the spinal cord. The clinical presentation of X-ALD varies even in the same family, which suggests the presence of modifier genes or environmental factors (Berger et al, 1994; Turk et al, 2017). Current therapeutic options are restricted to bone marrow transplantation (Miller et al, 2011) and hematopoietic stem cell gene therapy (Cartier et al, 2009; Eichler et al, 2017), and are limited by a very narrow therapeutic window, which reinforces the need to develop additional therapies for this devastating disease.

The mouse model of X-ALD (Abcd1⁻ mice) develops axonopathy and locomotor impairment very late in life, at 20 months of age, resembling adrenomyeloneuropathy, the most frequent X-ALD phenotype (Pujol et al, 2002). The closest homolog Abcd2 exhibits overlapping metabolic functions (Fourcade et al, 2009) and has been postulated as modifier of the biochemical defect (Muneer et al, 2014). Double mutant Abcd1⁻/Abcd2⁻/⁻ mice develop a more severe, earlier onset axonopathy starting at 12 months of age, what makes them a more suitable model for therapeutic essays (Pujol et al, 2002; Mastroeni et al, 2009; Lopez-Erauskin et al, 2011; Morato et al, 2013, 2015; Launay et al, 2015, 2017). Using these mouse models and patients' samples, studies by our laboratory and others have revealed that VLCFA-induced oxidative stress is a critical, early pathogenic factor in X-ALD (Vargas et al, 2004; Powers et al, 2005; Fourcade et al, 2008, 2010; Hein et al, 2008; Lopez-Erauskin et al, 2011; Petrillo et al, 2013), although the exact mechanisms by which VLCFA-induced redox imbalance causes neurodegeneration in X-ALD remain unclear (Fourcade et al, 2015). Here, we examined whether the NRF2 antioxidant pathway could contribute to the increased oxidative damage detected in this disease, in both the Abcd1⁻ mouse model and the skin fibroblasts derived from X-ALD patients. We also treated X-ALD mouse models (Abcd1⁻ and Abcd1⁻/Abcd2⁻/⁻ mice) with dimethyl fumarate (DMF, BG-12, Tecfidera), an NRF2 activator (Linker et al, 2011; Scannevin et al, 2012), that is a currently approved medication for relapsing-remitting multiple sclerosis (Fox et al, 2012; Gold et al, 2012).

# Results

### GSK-3β/NRF2 antioxidant pathway is altered in Abcd1⁻ mice

We previously identified a redox dyshomeostasis in X-ALD, characterized by an excess of reactive oxygen species (ROS) production and repression of key antioxidant enzymes (Fourcade et al, 2008). Since NRF2 plays a critical role in the antioxidant cellular defence,

we asked whether the NRF2-dependent antioxidant pathway was altered in the Abcd1 null mouse. We found decreased NRF2 protein levels in Abcd1⁻ mice spinal cord at 12 months of age (Fig 1A), a presymptomatic disease stage in this mouse model. Dysregulated NRF2 protein levels were organ-specific, as we did not observe any changes in non-affected tissues in the mouse model, such as cerebral cortex or liver (Fig EV1). To verify that lower protein levels had functional consequences, we measured mRNA expression of NRF2 classical target genes (Hmox1, Nqo1 and glutathione S-transferase alpha-3, Gsta3) at the same age. We observed a slight but significant decreased expression of these three NRF2 target genes in the Abcd1⁻ mouse spinal cord at 12 months of age (Fig 1B), consistent with a downregulated NRF2 pathway.

Several signals can regulate NRF2-dependent responses, in particular those that modulate GSK-3β activity (Salazar et al, 2006; Rojo et al, 2008; Rada et al, 2011). We thus examined the activity of the AKT/GSK-3β pathway in the spinal cord of Abcd1⁻ mice by measuring the phosphorylation of serine 473 (pSer473) and threonine 308 (pThr308) residues of AKT, which reflects its activation. We also measured the phosphorylation of serine 9 (pSer9) and tyrosine 216 (pTyr216) residues of GSK-3β, which indicate inhibition or activation of GSK-3β, respectively. We found less AKT activation in Abcd1⁻ mice spinal cord, as shown by decreased pThr308 AKT relative to total AKT levels. Defective AKT phosphorylation resulted in the activation of GSK-3β, indicated by reduced pSer9 GSK-3β compared with total GSK-3β levels (Fig 1C and D). We did not observe any changes in pSer473, pTyr216, or in the total levels of AKT and GSK-3β (Fig 1C and D). These data indicate a dysregulated AKT/GSK-3β/NRF2 axis in the Abcd1⁻ mouse spinal cord, with predicted higher activity of GSK-3β upstream of NRF2.

### Impaired NRF2-dependent antioxidant pathway is mediated by GSK-3β in patients' fibroblasts

Primary fibroblasts from X-ALD patients provide a good surrogate cell model to dissect disease mechanisms, as they recapitulate the main disease hallmarks: accumulation of VLCFA (Moser et al, 1980), higher production of free radicals of mitochondrial origin (Lopez-Erauskin et al, 2013), loss of energetic homeostasis (Galino et al, 2011), altered proteostasis (Launay et al, 2013, 2015), and endoplasmic reticulum (ER) stress (van de Beek et al, 2017; Launay et al, 2017). Using this cell system, we determined whether patients' fibroblasts exhibited an altered AKT/GSK-3β/NRF2 pathway.

At baseline, we observed equivalent NRF2 protein levels in patients' fibroblasts compared with controls (Fig EV1).

We then tested the functionality of the NRF2 pathway, by treating patients' and control fibroblasts either with C26:0, the primary VLCFA accumulated in patients, or with oligomycin, which acts as a generator of mitochondrial ROS inhibiting complex V (Fourcade et al, 2008; Paupe et al, 2009). Both compounds produce mitochondrial ROS in these fibroblasts (Lopez-Erauskin et al, 2013). We show that both C26:0 and oligomycin activated NRF2-dependent responses in control fibroblasts, characterized by both higher NRF2 translocation to the nucleus (Fig 2A and B) and increased expression of NRF2 target genes (HMOX1, NQO1, and GCLC mRNA; Fig 2C). However, this physiological response against oxidative stress was blunted in X-ALD fibroblasts with both ROS-producing stimuli (Fig 2A–C).

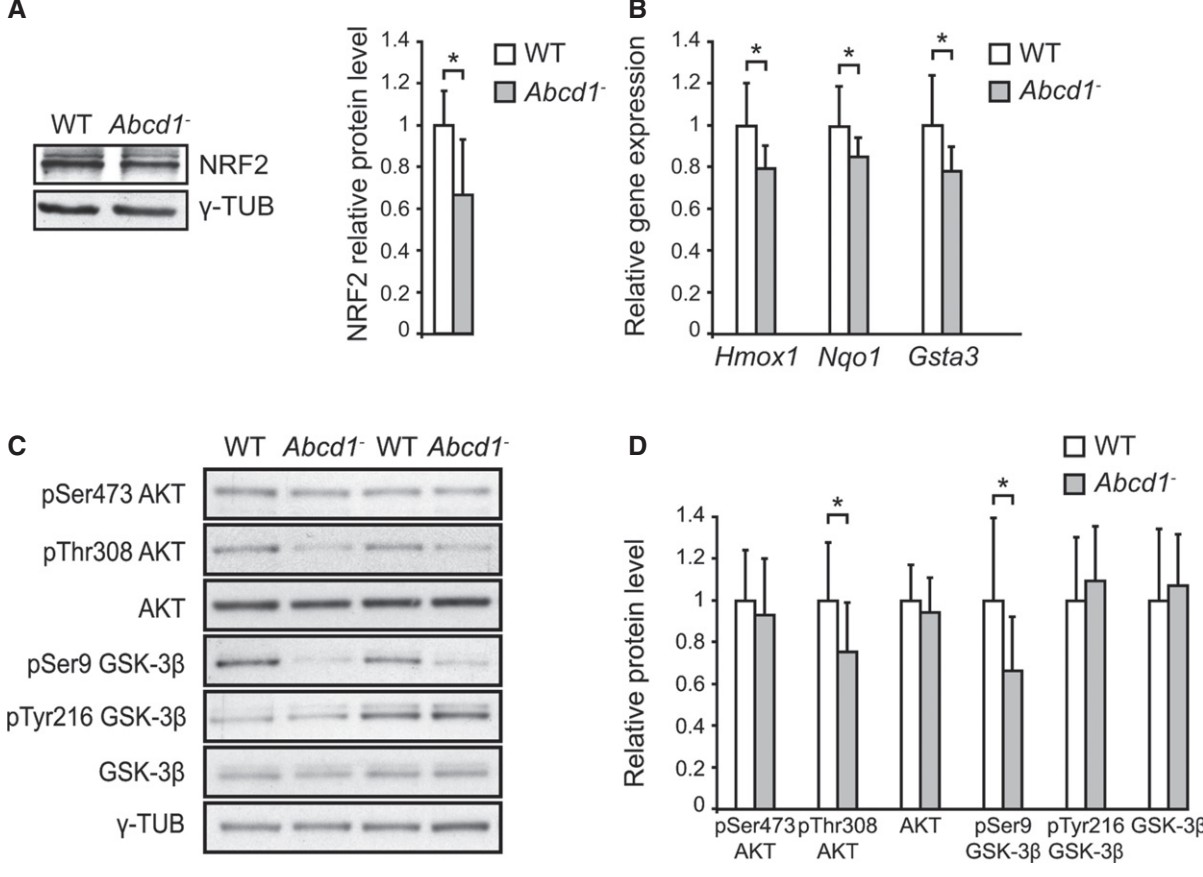

**Figure 1.  Altered GSK-3β/NRF2 antioxidant pathway in *Abcd1⁻* mice.**

A    Representative immunoblot of NRF2 protein level measured in WT ($n = 6$) and *Abcd1⁻* ($n = 6$) mice spinal cord at 12 months of age. Protein levels normalized relative to γ-tubulin (γ-TUB) and quantification depicted as fold change to WT mice.

B    NRF2-dependent antioxidant gene expression (*Hmox1*, *Nqo1*, and *Gsta3*) in WT ($n = 8$) and *Abcd1⁻* ($n = 8$) mice spinal cord at 12 months of age. Gene expression normalized relative to mouse *RplpO* and depicted as fold change to WT mice.

C, D    Representative immunoblots of pSer473 AKT, pThr308 AKT, AKT, pSer9 GSK-3β, pTyr216 GSK-3β, and GSK-3β protein level in WT ($n = 12$) and *Abcd1⁻* ($n = 12$) mice spinal cord at 12 months of age**.** Protein level normalized relative to corresponding non-phosphorylated proteins or γ-TUB (in the case of AKT and GSK-3β). Quantification depicted as fold change to WT mice.

Data information: In (A, B, and D), data are presented as mean $\pm$ SD. *$P < 0.05$ (unpaired Student's *t*-test). See the exact *P*-values in Appendix Table S3.
Source data are available online for this figure.

Moreover, both treatments elicited AKT activation (increased pSer473 and pThr308) and subsequent GSK-3β inactivation (higher pSer9 GSK-3β levels) in control fibroblasts (Fig 2D and E). Again, this physiological response against oxidative stress was impaired in X-ALD fibroblasts, as phosphorylated levels of AKT and GSK-3β did not change following C26:0 or oligomycin treatment (Fig 2D and E).

As GSK-3β activation can repress NRF2, we sought to determine whether this phenomenon was interrelated in the cellular model. For this, we assessed whether specific GSK-3β inhibitors (CT99021 and SB216763; Coghlan *et al*, 2000; Ring *et al*, 2003) could restore a normal NRF2-dependent response in X-ALD fibroblasts. Indeed, treatment with both compounds reactivated the NRF2 pathway, characterized by an upregulation of the NRF2-target genes *HMOX1*, *NQO1,* and *GCLC* in patients' fibroblasts upon incubation with excess of C26:0 (Fig 2F).

Collectively, these data indicate that the aberrant GSK-3β activation upstream of NRF2 governs the blunted NRF2-dependent response upon oxidative stress in this disease model.

**DMF rescues mitochondrial depletion, bioenergetic failure, and oxidative damage in *Abcd1⁻* mice**

To elucidate the impact of a defective NRF2-dependent response in the pathogenesis of adrenoleukodystrophy, we decided to treat *Abcd1⁻* mice with DMF, a classical activator of NRF2 (Linker *et al*, 2011; Scannevin *et al*, 2012). Dimethyl fumarate has therapeutic efficacy for relapsing-remitting multiple sclerosis (Fox *et al*, 2012; Gold *et al*, 2012) and besides, preclinical tests show success to treat other neurodegenerative diseases like Huntington's (Ellrichmann *et al*, 2011) and Parkinson's disease (Ahuja *et al*, 2016; Lastres-Becker *et al*, 2016).

Before treating the animals, we tested DMF in control and X-ALD fibroblasts. We found that DMF reactivated the NRF2-blunted response upon VLCFA addition (Fig EV2), similar to the GSK-3β inhibitors used (Fig 2F). Moreover, DMF alone induced *HMOX1* and *NQO1* expression in control fibroblasts and also *HMOX1* expression

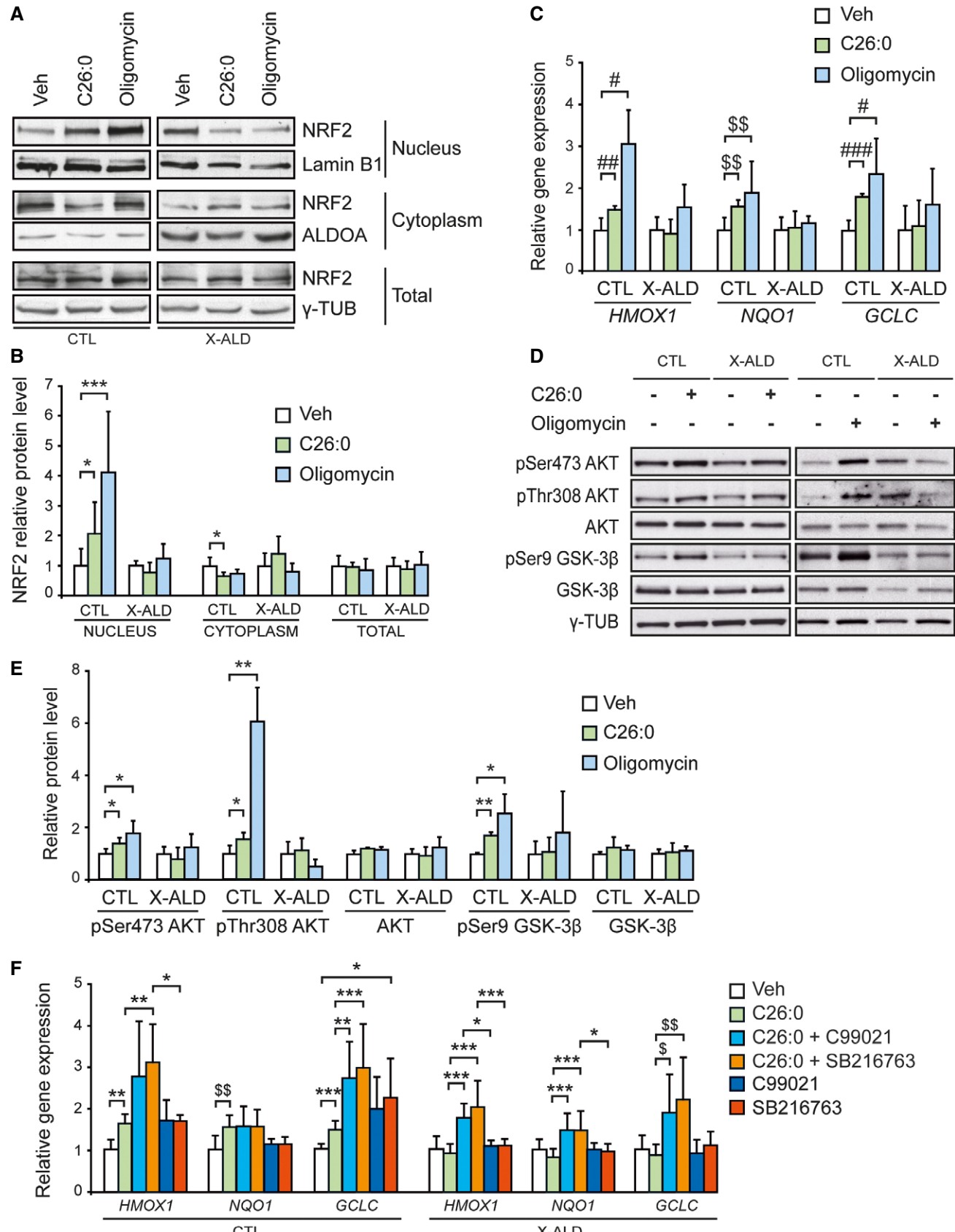

Figure 2.

**Figure 2.   Impaired AKT/GSK-3β/NRF2 antioxidant response after oxidative stress in X-ALD patients' fibroblasts.**

A, B   Representative immunoblots of NRF2 protein translocation to the nucleus upon VLCFA (C26:0, 50 μM, 24 h) or oligomycin (15 μM, 18 h) in control (CTL, *n* = 5 per condition, left panels) and X-ALD (*n* = 5 per condition, right panels) fibroblasts. Protein levels normalized relative to lamin B1 in the nuclear fraction, aldolase A (ALDOA) in the cytoplasmic fraction, and γ-TUB in the total fraction. Quantification depicted as fold change to vehicle-treated (Veh) fibroblasts.

C   NRF2-dependent antioxidant gene expression (*HMOX1*, *NQO1*, and *GCLC*) upon oxidative stress in CTL (*n* = 5 per condition) and X-ALD (*n* = 5 per condition) fibroblasts. Gene expression normalized relative to *RPLP0*. Quantification depicted as fold change to vehicle-treated (Veh) fibroblasts.

D, E   Representative immunoblots of pSer473 AKT, pThr308 AKT, AKT, pSer9 GSK-3β, and GSK-3β measured after oxidative stress in CTL (*n* = 5 per condition) and X-ALD (*n* = 5 per condition) fibroblasts. Protein levels normalized relative to corresponding non-phosphorylated proteins or γ-TUB (in the case of AKT and GSK-3β). Quantification depicted as fold change to vehicle-treated (Veh) fibroblasts.

F   NRF2-dependent antioxidant gene expression (*HMOX1*, *NQO1*, and *GCLC*) after GSK-3β inhibition in VLCFA-treated CTL (*n* = 8 per condition) and X-ALD (*n* = 8 per condition) fibroblasts. Gene expression normalized relative to *RPLP0*. Quantification depicted as fold change to vehicle-treated (Veh) fibroblasts.

Data information: In (B, C, E, and F), data are presented as mean ± SD. In (B, E, and F), *$^*P < 0.05$, $^{**}P < 0.01$, $^{***}P < 0.001$ (one-way ANOVA followed by Tukey's *post hoc* test). In (C), $^#P < 0.05$, $^{##}P < 0.01$, $^{###}P < 0.001$ (one-way ANOVA followed by Dunnett's *post hoc* test). In (C and F), $^$P < 0.01$, $^{$$}P < 0.01$ (non-parametric Kruskal–Wallis' test followed by Dunn's *post hoc* test). See the exact *P*-values in Appendix Table S3.
Source data are available online for this figure.

in X-ALD fibroblasts (Fig EV2). Thus, these new data reinforced the rational for DMF treatment *in vivo*.

We fed *Abcd1⁻* mice with DMF-containing chow at 100 mg/kg, starting at 8 months of age, for 4 months. First, we verified the efficacy of dietary DMF administration by measuring NRF2 protein levels and mRNA expression of three classical NRF2-target genes (*Hmox1*, *Nqo1*, and *Gsta3*). Dimethyl fumarate treatment rescued both NRF2 protein levels (Fig 3A) and NRF2 targets in *Abcd1⁻* mice spinal cord at 12 months of age (Fig 3B).

Next, we measured the effect of DMF on several quantitative markers of oxidative damage to lipids and proteins, such as direct carbonylation of proteins (Aminoadipic semialdehyde: AASA), glycoxidation (Nε-(carboxyethyl)-lysine: CEL and Nε-(carboxymethyl)-lysine: CML), and protein lipoxidation (Nε-malondialdehyde-lysine: MDAL; Fourcade *et al*, 2008). We found an antioxidant role for DMF in this model, as it normalized AASA, CEL, CML, and MDAL in *Abcd1⁻* mice spinal cord (Fig 3C).

We also examined the effect of DMF on mitochondrial dysfunction (Morato *et al*, 2013, 2015). DMF normalized mitochondrial biogenesis, based on different parameters: mtDNA levels (Fig 3D) and mRNA expression of sirtuin-1, *Sirt1*; peroxisome proliferator-activated receptor gamma coactivator 1-alpha, *Ppargc1a*; nuclear respiratory factor-1, *Nrf1*; and transcription factor A, mitochondrial, *Tfam* (Fig 3E; Morato *et al*, 2013, 2015). We previously reported decreased levels of ATP in the spinal cord of *Abcd1⁻* mice (Galino *et al*, 2011), suggesting that deficient energy homeostasis is a key feature in X-ALD pathology. In this study, we reveal that DMF prevented bioenergetic failure, as it normalized ATP levels (Fig 3F). These effects seem to be independent of VLCFA levels, since DMF treatment did not alter C24:0 or C26:0 levels in the spinal cord of 12-month-old *Abcd1⁻* mice (Fig EV3). Altogether, DMF activated NRF2-dependent antioxidant pathway and prevented mitochondrial depletion, bioenergetic failure, and oxidative damage in the spinal cord of the disease mouse model.

**DMF treatment prevents inflammatory imbalance in *Abcd1⁻* mice**

Although patients presenting with pure adrenomyeloneuropathy do not exhibit overt brain inflammation that induces demyelination, we previously found low-grade inflammatory dysregulation in the *Abcd1⁻* mouse spinal cord and in adrenomyeloneuropathy patients. Our functional genomics assay detected activation of the NF-κB-mediated inflammatory pathway and increased expression of

several pro-inflammatory cytokines in the *Abcd1⁻* mouse spinal cord (Schluter *et al*, 2012). In adrenomyeloneuropathy patients, we recently reported a general dysregulation of inflammatory pathways in peripheral blood mononuclear cells (PBMC) and plasma (Ruiz *et al*, 2015).

Since DMF is also a classical immunomodulatory drug (Schilling *et al*, 2006; Linker *et al*, 2011), we examined its effects on mRNA expression of several inflammation-related genes in the *Abcd1⁻* mice spinal cord. At 12 months of age, *Abcd1⁻* mice exhibited a general imbalance of both pro- and anti-inflammatory markers, characterized by the induction of pro-inflammatory markers including nuclear factor kappa B subunit 2 (*Nfkb2*), interleukin 1 beta (*Il1b*), tumor necrosis factor alpha (*Tnfa*), tumor necrosis factor receptor superfamily member 1a (*Tnfrsf1a*), chemokine (C-C motif) ligand 5 (*Ccl5*), chemokine (C-X-C motif) ligand 9 (*Cxcl9*), chemokine (C-X-C motif) ligand 10 (*Cxcl10*), and chemokine (C-C motif) receptor type 6 (*Ccr6*) (Fig 4A). Also, we observed an upregulation of some anti-inflammatory markers, such as chitinase-like 3 (*Chil3*), chemokine (C-X-C motif) ligand 12 (*Cxcl12*), insulin-like growth factor 1 (*Igf1*), and transforming growth factor, beta 1 (*Tgfb1*) (Fig 4B). Interleukin 6 (*Il6*), resistin like alpha (*Retnla*, also called *Fizz1*), and macrophage migration inhibitory factor (*Mif*) were decreased in the *Abcd1⁻* mouse spinal cord (Fig 4A and B).

Dimethyl fumarate prevented most alterations observed in the inflammatory profile in the *Abcd1⁻* mice spinal cord and normalized mRNA levels of *Nfkb2*, *Il6*, *Tnfa*, *Ccl5*, *Cxcl10*, *Ccr6* (pro-inflammatory; Fig 4A), and *Mif*, *Cxcl12*, *Tgfb1*, *Igf1* (anti-inflammatory; Fig 4B). However, DMF had no effect on *Tnfrsf1a*, *Cxcl9* (pro-inflammatory), and *Fizz1* (anti-inflammatory) mRNA expression in the *Abcd1⁻* mice spinal cord (Fig 4A and B). DMF exacerbated the induction of *Il1b* (pro-inflammatory) and *Chil3* (anti-inflammatory), and induced the expression of interleukin 10 (*Il10*), an anti-inflammatory cytokine, in the *Abcd1⁻* mouse spinal cord (Fig 4A and B). These data demonstrate that DMF normalized the inflammatory profile in *Abcd1⁻* mice.

**DMF halts axonal degeneration in *Abcd1⁻/Abcd2⁻/⁻* mice**

We then evaluated the effects of DMF on axonal degeneration and locomotor impairment in X-ALD mouse model. We fed *Abcd1⁻/ Abcd2⁻/⁻* (DKO) mice with DMF for 6 months, starting at 12 months of age. First, we characterized the immunohistochemical signs of neuropathology present in DKO mice at 18 months of age,

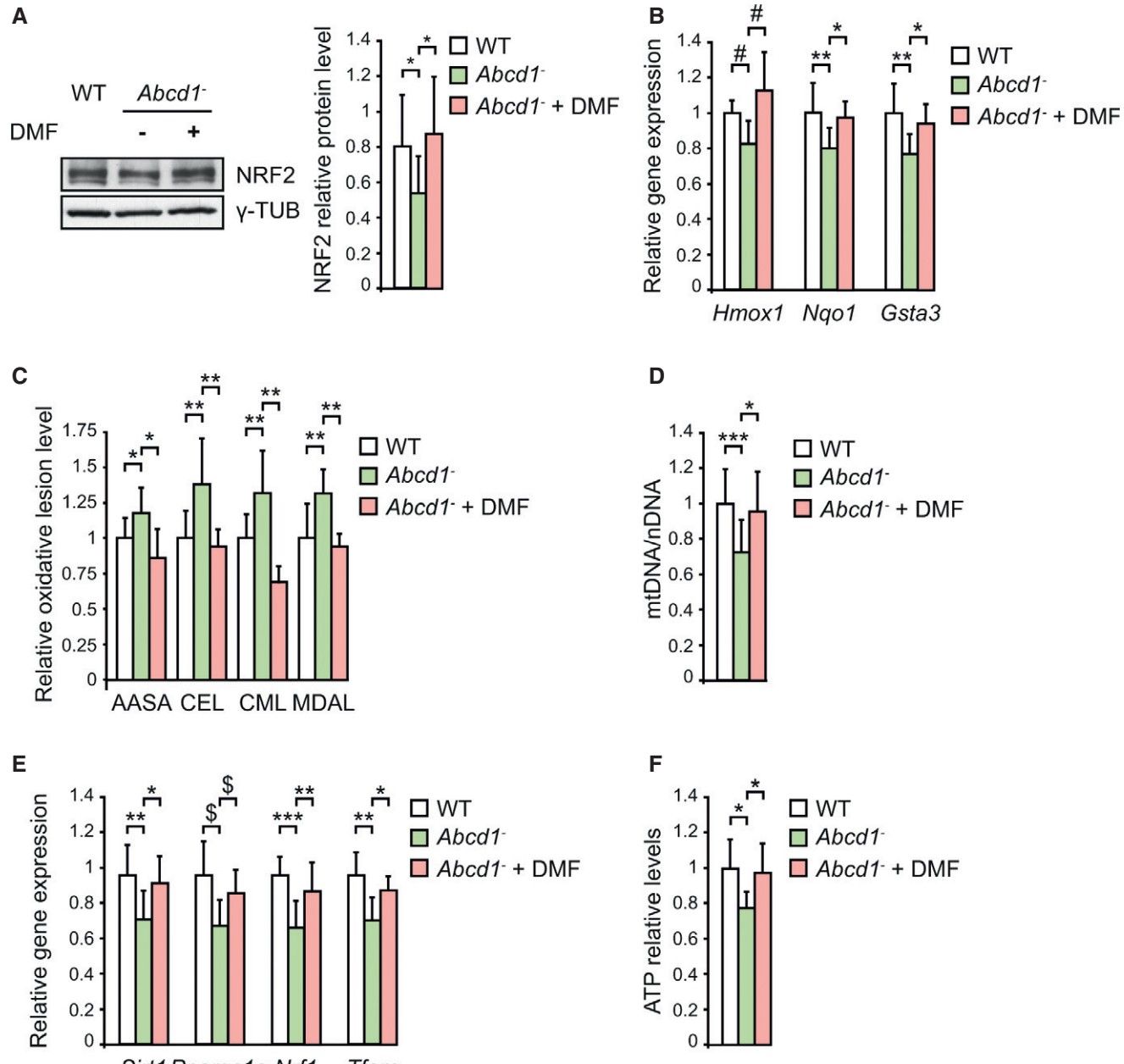

**Figure 3. NRF2 activation by DMF prevents oxidative damage to proteins and lipids, mitochondrial depletion, and bioenergetic failure in *Abcd1⁻* mice.**

A   Representative immunoblot of NRF2 protein levels in WT ($n = 6$), *Abcd1⁻* ($n = 6$) and DMF-treated *Abcd1⁻* mice (*Abcd1⁻* + DMF, $n = 6$) mice spinal cord at 12 months of age. Protein levels normalized relative to γ-TUB. Quantification depicted as fold change to WT mice.

B   NRF2-dependent antioxidant gene expression (*Hmox1, Nqo1,* and *Gsta3*) in WT ($n = 8$), *Abcd1⁻* ($n = 8$), and *Abcd1⁻* + DMF ($n = 8$) mice spinal cord at 12 months of age. Gene expression normalized relative to *Rplp0*. Quantification represented as fold change to WT mice.

C   Oxidative lesions to lipids and proteins in WT ($n = 5$), *Abcd1⁻* ($n = 5$), and *Abcd1⁻* + DMF ($n = 5$) mice spinal cord at 12 months of age. AASA, CEL, CML, and MDAL levels measured by GC/MS. Quantification represented as fold change to WT mice.

D   mtDNA levels in WT ($n = 8$), *Abcd1⁻* ($n = 8$), and *Abcd1⁻* + DMF ($n = 8$) mice spinal cord at 12 months of age. mtDNA content expressed as the ratio of mtDNA (*CytB* levels) to nDNA (*Cebpa* levels). Quantification depicted as fold change to WT mice.

E   *Sirt1, Ppargc1a, Nrf1,* and *Tfam* gene expression in WT ($n = 8$), *Abcd1⁻* ($n = 8$), and *Abcd1⁻* + DMF ($n = 8$) mice spinal cord at 12 months of age. Gene expression normalized relative to *Rplp0*. Quantification depicted as fold change to WT mice.

F   ATP levels in WT ($n = 8$), *Abcd1⁻* ($n = 8$), and *Abcd1⁻* + DMF ($n = 8$) mice spinal cord at 12 months of age. Quantification represented as fold change to WT mice.

Data information: Data are presented as mean ± SD. *$P < 0.05$, **$P < 0.01$, ***$P < 0.001$ (one-way ANOVA followed by Tukey's *post hoc* test). In (B), #$P < 0.05$ (one-way ANOVA followed by Dunnett's *post hoc* test). In (E), $$P < 0.05$ (non-parametric Kruskal–Wallis' test followed by Dunn's *post hoc* test). See the exact *P*-values in Appendix Table S3.

Source data are available online for this figure.

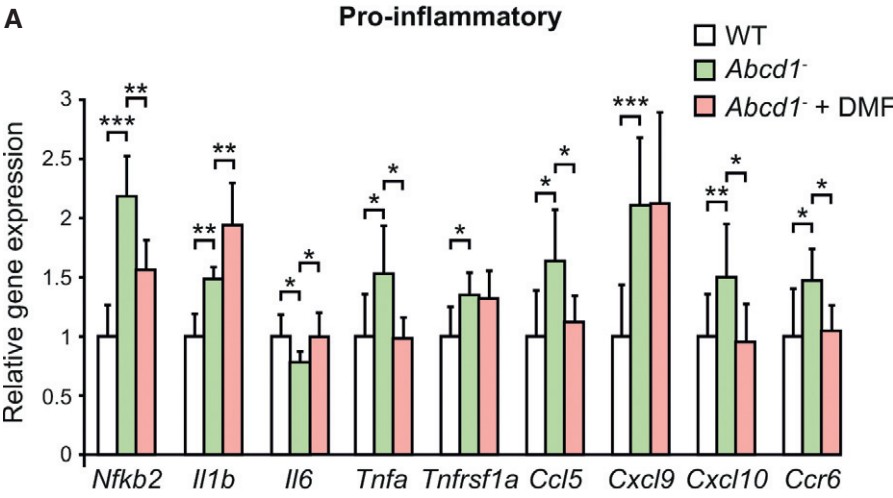

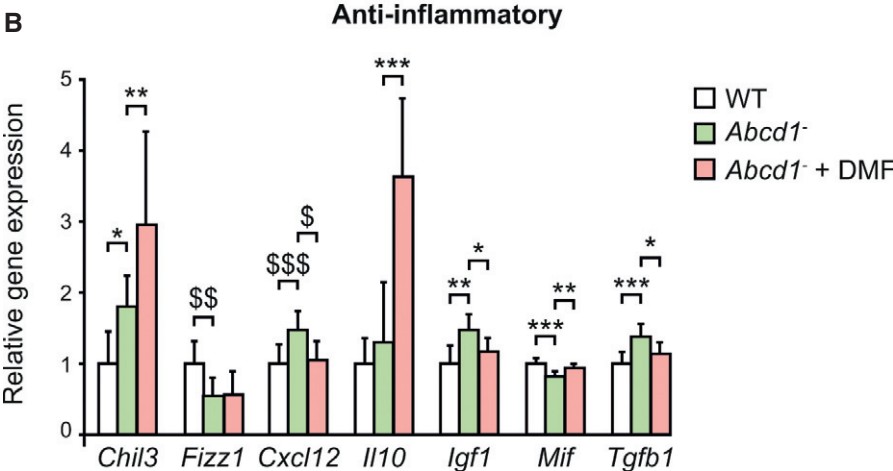

**Figure 4. DMF treatment restores inflammatory profile in *Abcd1⁻* mice.**

A    Pro-inflammatory (*Nfkb2, Il1b, Il6, Tnfa, Tnfrsf1a, Ccl5, Cxcl9, Cxcl10, Ccr6*) gene expression profile.
B    Anti-inflammatory (*Fizz1, Chil3, Cxcl12, Il10, Igf1, Mif, Tgfb1*) gene expression profile.

Data information: Gene expression was measured in WT (*n* = 8), *Abcd1⁻* (*n* = 8), and *Abcd1⁻* + DMF (*n* = 8) mice spinal cord at 12 months of age. Expression of cytokines, chemokines, and other inflammation-related genes was normalized relative to *Rplp0*. Quantification depicted as fold change to WT mice. Data are presented as mean ± SD. In (A and B), \*$P < 0.05$, \*\*$P < 0.01$, \*\*\*$P < 0.001$ (one-way ANOVA followed by Tukey's *post hoc* test). In (B), $^\$P < 0.05$, $^{\$\$}P < 0.01$, $^{\$\$\$}P < 0.001$ (non-parametric Kruskal–Wallis' test followed by Dunn's *post hoc* test). See the exact *P*-values in Appendix Table S3.

after DMF treatment. We assessed (i) microgliosis; (ii) astrocytosis; (iii) axonal degeneration, shown by accumulation of amyloid precursor protein (APP) and synaptophysin in axonal swellings; (iv) lipidic myelin debris, shown by Sudan Black staining (Pujol *et al*, 2004); (v) oxidative damage to DNA, indicated by increased 8-oxo-7,8-dihydro-2′-deoxyguanosine (8-oxo-dG) staining (Lopez-Erauskin *et al*, 2011); (vi) unhealthy motor neurons with reduced staining of SMI-32, an antibody that labels a non-phosphorylated epitope of neurofilament proteins; and (vii) decreased mitochondrial content observed by cytochrome c (Cyt C) staining in motor neurons (Morato *et al*, 2013; Fig 5A–Y). Dimethyl fumarate reversed microgliosis and astrocytosis, as it normalized the density of astrocytes and microglial cells in DKO mice (Fig 5A–F and Y), prevented axonal accumulation of APP and synaptophysin (Fig 5G–L and Y), halted the appearance of myelin debris along the spinal cord

(Fig 5M–O), and reduced DNA oxidative damage shown by 8-oxo-dG staining (Fig 5P–R) in DKO mice. In addition, motor neuron health and mitochondrial levels improved with DMF treatment (Fig 5S–X). Altogether, these data reveal that DMF treatment significantly ameliorated the neuropathology in *Abcd1⁻/Abcd2⁻/⁻* mice.

### DMF reverses locomotor deficits in *Abcd1⁻/Abcd2⁻/⁻* mice

Next, we measured the effect of DMF on the locomotor phenotype of DKO mice using bar cross and treadmill tests at the end of the treatment. As previously described (Pujol *et al*, 2004; Lopez-Erauskin *et al*, 2011), DKO mice at 18 months of age took longer time to cross the bar and slipped off more times while crossing the bar. However, DMF-treated DKO mice behaved similar to wild-type (WT) mice. These data indicate that DMF improved the ability of DKO mice to cross the bar

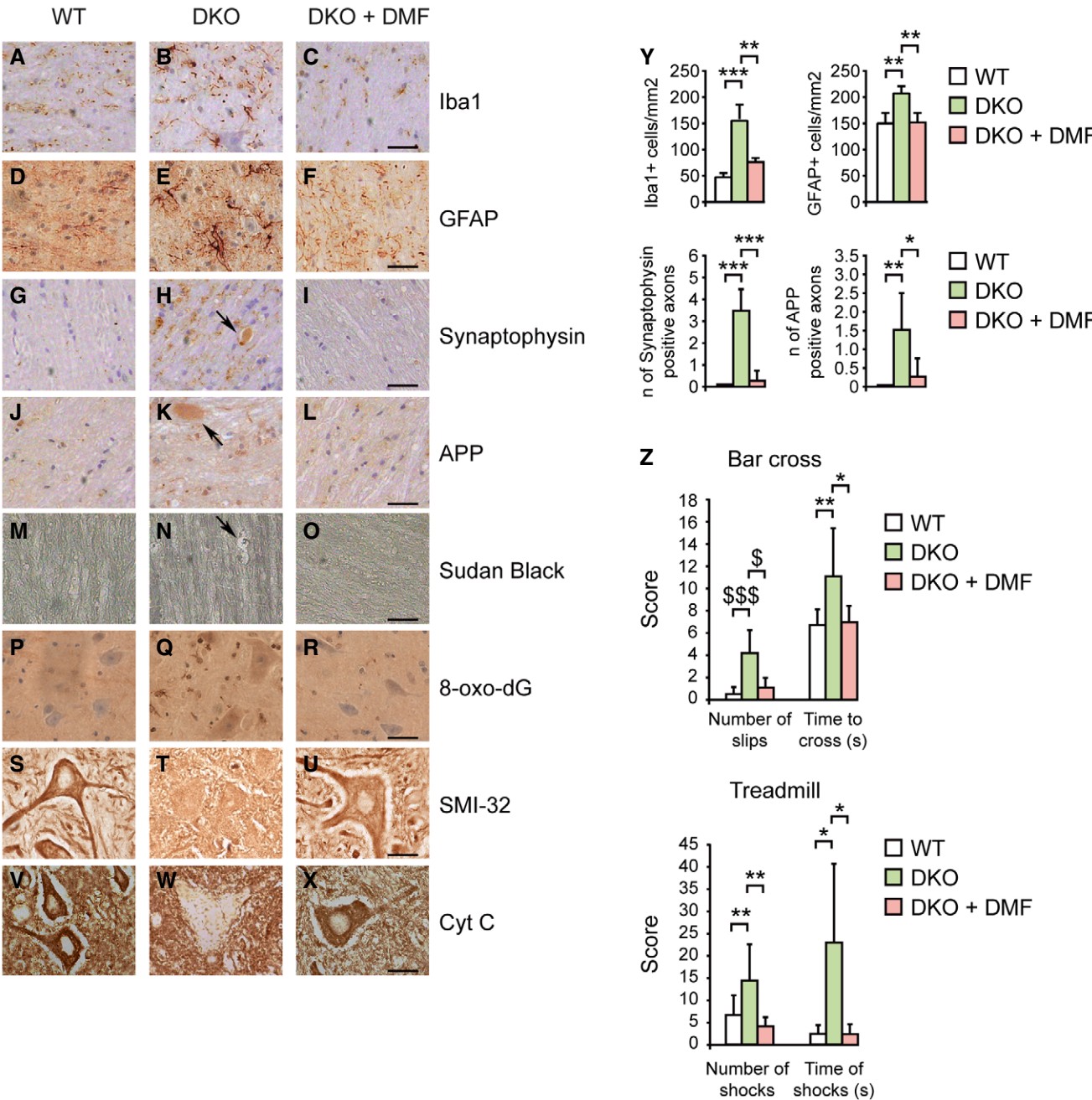

**Figure 5.  DMF treatment halts axonal degeneration and locomotor impairment in *Abcd1⁻/Abcd2⁻/⁻* mice.**

A–X  Immunohistological analysis of axonal pathologies performed in 18-month-old WT, *Abcd1⁻/Abcd2⁻/⁻* (DKO), and DKO mice treated with DMF (DKO + DMF) (n = 5 per genotype and condition). Spinal cord immunohistological sections processed for (A–C) Iba1, (D–F) GFAP, (G–I) synaptophysin, (J–L) APP, (M–O) Sudan Black, (P–R) 8-oxo-dG, (S–U) SMI-32, and (V–X) Cyt C. Representative images for (A, D, G, J, M, P, S, V) WT, (B, E, H, K, N, Q, T, W) DKO, and (C, F, I, L, O, R, U, X) DKO + DMF are shown. Scale bars = 125 μm (A–R) and 25 μm (S–X). Black arrows indicate APP or synaptophysin accumulations (H and K), and myelin debris (N) in DKO mice.

Y   Quantification of GFAP⁺ and Iba1⁺ cells per mm², and synaptophysin and APP accumulations in spinal cord immunohistological sections of WT (n = 5), DKO (n = 5), and DKO + DMF (n = 5) mice.

Z   Bar cross and treadmill tests performed on 18-month-old WT (n = 14), DKO (n = 16), and DKO + DMF (n = 14) mice. In the bar cross, data refer to number of slips and time (seconds) spent to cross the bar. In the treadmill test, data refer to number of shocks and time of shocks at the last time point measured (7 min and 30 s).

Data information: In (Y and Z), data are presented as mean ± SD. In (Y and Z), *P < 0.05, **P < 0.01, ***P < 0.001 (one-way ANOVA followed by Tukey's *post hoc* test), $P < 0.05, $$$P < 0.001 (non-parametric Kruskal–Wallis' test followed by Dunn's *post hoc* test). See the exact *P*-values in Appendix Table S3.

(Fig 5Z). As earlier described (Morato et al, 2015), DKO mice behaved worse than WT in the treadmill test, as the total number and duration of shocks were higher than in WT. Dimethyl fumarate treatment also ameliorated the performance of DKO mice in this test (Fig 5Z). In summary, these data indicate that DMF treatment halted the progression of the locomotor deficits observed in $Abcd1^-/Abcd2^{-/-}$ mice.

# Discussion

In this work, we uncover an impaired AKT/GSK-3β/NRF2 axis in X-ALD, composed of a blunted NRF2-dependent response which obeys an aberrant upstream activation of GSK-3β. This impaired NRF2-dependent antioxidant response suggests a mechanism by which excess of C26:0 causes oxidative damage only in patients' fibroblasts in contrast with controls, even though both populations present similar amounts of ROS after C26:0 treatment (Fourcade et al, 2008). Indeed, the NRF2 target genes Hmox1, Nqo1, and Gsta3, whose expression we show is reduced in $Abcd1^-$ mice spinal cord, mediate a cellular defence against toxic, carcinogenic, and pharmacologically active electrophilic compounds (Lee et al, 2003; Dinkova-Kostova & Talalay, 2010). We propose that oxidative damage in this disease may be a consequence of the low activity of these enzymes, incapable of compensating for the increased ROS production. Inactivation of the NRF2 pathway is not a phenomenon limited to X-ALD, but is reported in several neurodegenerative disorders including Alzheimer's disease (Ramsey et al, 2007; Kanninen et al, 2008), amyotrophic lateral sclerosis (Sarlette et al, 2008), Friedrich's ataxia (Paupe et al, 2009; Shan et al, 2013), and experimental autoimmune encephalomyelitis mouse models (Morales Pantoja et al, 2016). A recent meta-analysis identified the NRF2 pathway as a common dysregulated hub in Alzheimer's and Parkinson's disease patients (Wang et al, 2017). Moreover, NRF2 can contribute to Parkinson's and Huntington's disease pathology (Johnson & Johnson, 2015). Indeed, NRF2-deficient mice are more vulnerable to striatal toxicity induced by systemic administration of 3-nitropropionic acid (3-NP) (Calkins et al, 2005), 1-methyl-4-phenyl-1,2,3,6 tetrahydropyridine (MPTP) (Burton et al, 2006; Chen et al, 2009), or 6-hydroxydopamine (6-OHDA) used to induce basal ganglia neural dysfunction (Jakel et al, 2007). Of note, NRF2-deficient mice also develop a more severe myelin oligodendrocyte glycoprotein (MOG)-induced experimental autoimmune encephalomyelitis with increased oxidative damage in the CNS, finally leading to enhanced demyelination and more pronounced axonal loss (Johnson et al, 2010). This provides a direct link between insufficient antioxidant response and axonal damage. Further, genetic variations in the NRF2 gene have been associated with risk and/or age of onset in amyotrophic lateral sclerosis, Alzheimer's, and Parkinson's disease (von Otter et al, 2010a,b; Bergstrom et al, 2014).

Glial cells are involved in the NRF2-mediated neuroprotective effects. Ex vivo studies and analysis of neurodegenerative models for motor neuron disorders, Parkinson's disease, or cerebral ischemia indicate that NRF2-mediated neuroprotection critically involves astrocyte-induced effects (Kraft et al, 2004; Shih et al, 2005; Chen et al, 2009). Initial evidence of the importance of NRF2 in glial cells comes from Nrf2 knockout mice, which display astrogliosis and myelinopathy in the cerebellum (Hubbs et al, 2007). Also, striatum is protected from MPTP toxicity in transgenic mice which overexpress NRF2 in astrocytes (Chen et al, 2009).

Little is known about the link between AKT/GSK-3β and NRF2 in neurodegenerative diseases. Exacerbated GSK-3β activity is also present in other peroxisomal diseases, like rhizomelic chondrodysplasia punctata, in which plasmalogen deficiency leads to AKT inactivation and GSK-3β activation (da Silva et al, 2014). Future work should address and quantify both the functionality of the NRF2 pathway in rhizomelic chondrodysplasia punctata and the AKT/GSK-3β/NRF2 axis in other peroxisomal disorders, given the possibility of targeted treatments like DMF, for these metabolic diseases. Increased GSK-3β also occurs in other neurodegenerative conditions, like tauopathies, in which GSK-3β is one of the kinases responsible for pathological phosphorylation of Tau (Llorens-Martin et al, 2014); Parkinson's disease (Duka et al, 2009; Credle et al, 2015) or multiple sclerosis (Beurel et al, 2013).

In the present work, we inhibited GSK-3β in X-ALD fibroblasts using two different drugs, which restored NRF2 pathway function and activated transcription of NRF2 target genes upon oxidative stress. Thus, we reveal GSK-3β inhibition as a new therapeutic strategy in X-ALD, which is currently under investigation for other neurodegenerative diseases. In the Senescence Accelerated Mouse-Prone 8 (SAMP8) mouse, an Alzheimer's disease mouse model, GSK-3β inhibition caused NRF2 activation and decreased oxidative stress, together with reduced Tau phosphorylation and improved learning and memory (Farr et al, 2014). Another study uncovered the therapeutic potential of GSK-3β pathway inhibition to restore neurodevelopmental defects in hereditary spastic paraplegia (HSP) patients with SPG11 mutations (Mishra et al, 2016). Pathways that modulate GSK-3β, such as phosphoinositide 3-kinase (PI3K)/AKT and WNT/β-catenin, also regulate myelination (Fancy et al, 2009; Guo et al, 2016). Unfortunately, GSK-3 inhibitors have so far met little success in clinical trials for neurodegenerative diseases. Two phase II clinical trials with the GSK-3β inhibitor, Tideglusib, showed no clinical benefits in Alzheimer's disease (Lovestone et al, 2015) and progressive supranuclear palsy (Tolosa et al, 2014). We therefore chose DMF to activate NRF2 in our X-ALD preclinical models using a similar dosage to that administered to human multiple sclerosis patients, paving the way to clinical translation. DMF treatment is advantageous because it enhances the NRF2 antioxidant pathway, but also exerts pleiotropic effects improving proteostasis, mitochondrial function, and neuroinflammation (Johnson & Johnson, 2015), cellular responses that also contribute to the pathogenesis of adrenoleukodystrophy (Fourcade et al, 2015). Indeed, previous in vitro studies described the effects of DMF on mitochondrial function (Scannevin et al, 2012; Ahuja et al, 2016; Peng et al, 2016). Here, we describe a positive effect of DMF on mitochondrial biogenesis and function in the central nervous system in vivo, characterized by increased mtDNA levels and mitochondrial biogenesis regulatory gene expression (Sirt1, Ppargc1a, Nrf1, Tfam), as well as normalized ATP levels in the spinal cord of $Abcd1^-$ mice. These results are supported by recent data in mice and multiple sclerosis patients treated with DMF (Hayashi et al, 2017).

Neuroinflammation is another common feature of neurodegenerative diseases. In this study, we observed normalization in the gene expression of Nfkb2 transcription factor and pro-inflammatory cytokines like Tnfa, Ccl5, Cxcl10, and Ccr6, concomitant with an increase in the anti-inflammatory cytokines Il10 and Chil3. This DMF effect is consistent with previous reports showing an in vivo anti-inflammatory effect in the experimental autoimmune encephalomyelitis animal model (Schilling et al, 2006). We also

show that DMF prevented microgliosis and astrocytosis in $Abcd1^-/Abcd2^{-/-}$ mice, consistent with results from recent studies in Parkinson's disease mouse models (Jing et al, 2015; Lastres-Becker et al, 2016). NRF2 activation also occurs in PBMC and glial cells from multiple sclerosis patients treated with DMF from the DEFINE and CONFIRM studies (Gopal et al, 2017). Yet, the immunomodulatory effect of DMF in the nervous system can be NRF2-dependent (Linker et al, 2011) or independent (Brennan et al, 2016). G protein-coupled receptor 109A (GPR109A), also known as the hydroxycarboxylic acid receptor 2 (HCA2), is another DMF target (Parodi et al, 2015). Future studies on the role of HCA2 in X-ALD and other demyelinating diseases will further enlighten the mechanism of action of DMF in the neuroinflammatory axis.

In view of these data, DMF exhibits a great potential to treat X-ALD and other neurodegenerative diseases with an overall good safety profile (Fox et al, 2012; Gold et al, 2012). However, some precautions need to be observed with this drug, as side effects need to be monitored and evaluated carefully. The most important is lymphocytopenia (Fox et al, 2016). Of > 230,000 patients treated with DMF in the period of 3 years following commercial availability, five cases of progressive multifocal leukoencephalopathy (PML) have been reported, in the setting of moderate to severe prolonged lymphocytopenia (Pardo & Jones, 2017). As a consequence, FDA and EMA (EMA/627077/2015) recently issued updated safety recommendations to minimize PML risk, which include regular blood counts. Recent reports indicate that a reduction in T cells rather than a general reduction in lymphocyte count may be associated with these cases (Gieselbach et al, 2017), which still represent a very low percentage of all the patients treated with fumaric acid esters.

In summary, our data uncover a novel role of GSK-3β/NRF2 in the physiopathogenesis of X-ALD. By identifying the mechanism impaired in the endogenous antioxidant response in this disease, we reveal a novel therapeutic intervention using DMF treatment to overcome the molecular pathogenesis and clinical signs of adrenoleukodystrophy in the mouse. Our data provide strong rationale to propose phase II clinical trials with DMF in adrenoleukodystrophy patients.

# Materials and Methods

### Reagents and antibodies

The following chemicals were used: DMF (Ref. 242926), hexacosanoic acid (C26:0, Ref. H0388), and oligomycin (Ref. O4876) were purchased from Sigma-Aldrich (Steinheim, Germany). GSK-3β inhibitors CHIR99021 (6-[[2-[[4-(2,4-dichlorophenyl)-5-(5-methyl-1H-imidazol-2-yl)-2-pyrimidinyl]amino]ethyl]amino]-3-pyridinecarbonitrile; Ref. 361559) and SB216763 (3-(2,4-dichlorophenyl)-4-(1-methyl-1H-indol-3-yl)-1H-pyrrole-2,5-dione; Ref. 1616) were purchased from Calbiochem (Billerica, MA, USA) and Tocris Biosciences (Bristol, UK), respectively. Detailed information on antibodies is summarized in Appendix Table S1.

### Mouse experiments

We used male mice of a pure C57BL/6J background. All methods employed in this study were in accordance with the ARRIVE guidelines, the Guide for the Care and Use of Laboratory Animals (Guide, 8th edition, 2011, NIH) and European (2010/63/UE) and Spanish (RD 53/2013) legislation. Experimental protocols were approved by IDIBELL, IACUC (Institutional Animal Care and Use Committee), and regional authority (3546 DMAH, Generalitat de Catalunya, Spain). IDIBELL animal facility is accredited by The Association for Assessment and Accreditation of Laboratory Animal Care (AAALAC, Unit 1155). Animals were housed at 22°C on specific pathogen-free conditions, in a 12-h light/dark cycle, and ad libitum access to food and water. Cages contained three to four animals.

We used two X-ALD mouse models in this study. We evaluated the biochemical signs of adult X-ALD in $Abcd1^-$ mice at 12 months of age. These mice present oxidative stress (Fourcade et al, 2008) and energetic homeostasis impairment (Galino et al, 2011) before the first clinical signs of adrenomyeloneuropathy-like pathology (axonopathy and locomotor impairment) appear at 20 months (Pujol et al, 2002).

To address the therapeutic effect of DMF, we assessed the clinical signs of adrenomyeloneuropathy in $Abcd1^-/Abcd2^{-/-}$ (DKO) mice, which display increased VLCFA accumulation in the spinal cord (Pujol et al, 2004), higher levels of oxidative damage to proteins (Fourcade et al, 2008; Galino et al, 2011), and a more severe adrenomyeloneuropathy-like pathology with an earlier onset at 12 months of age (Pujol et al, 2004). These mice are the preferred X-ALD mouse model for therapeutic testing (Mastroeni et al, 2009; Lopez-Erauskin et al, 2011; Morato et al, 2013, 2015; Launay et al, 2015, 2017).

For biochemical analysis, we euthanized the mice and stored the tissues at −80°C after snap-freezing them in liquid nitrogen. For histological analysis, spinal cord was harvested from 18-month-old mice after perfusing them with 4% paraformaldehyde (PFA; Sigma-Aldrich, Ref. 441244) in 0.1 M phosphate buffer pH 7.4. Histological and behavioral experiments were performed in a blind manner with respect to the animal's genotype and the treatment administered.

### DMF administration to mice

Dimethyl fumarate was mixed into AIN-76A chow from Dyets (Bethlehem, PA, USA) to provide a dose of 100 mg/kg/day. Human equivalent dose would be 8 mg/kg/day (240 mg in a typical 60 kg person). This is equivalent to the starting dose of BG-12/Tecfidera for multiple sclerosis patients, 120 mg twice a day (EMA/204830/2013).

To characterize biochemical signs in adult X-ALD mice, 8-month-old animals were randomly assigned to one of the following dietary groups for 4 months. Group I: WT mice received normal AIN-76A chow ($n = 12$); group II: $Abcd1^-$ mice received normal AIN-76A chow ($n = 12$); group III: $Abcd1^-$ mice received AIN-76A chow containing DMF ($n = 12$). To evaluate the effect of DMF on the clinical signs of adrenomyeloneuropathy-like pathology, 12-month-old animals were randomly assigned to one of the following dietary groups for 6 months. Group I: WT mice received normal AIN-76A chow ($n = 14$); group II: $Abcd1^-/Abcd2^{-/-}$ mice received normal AIN-76A chow ($n = 16$); and group III: $Abcd1^-/Abcd2^{-/-}$ mice received AIN-76A chow containing DMF ($n = 14$). DMF had no effect on weight or food intake under any treatment protocol, and none of the mice administered with DMF experienced any adverse events or death during the treatment.

## Human samples

Primary human fibroblasts were prepared from skin biopsies collected from healthy individuals ($n = 8$) and adrenomyeloneuropathy patients ($n = 8$) according to the IDIBELL guidelines for sampling, including informed consent obtained from the persons involved or their representatives according to the Declaration of Helsinki and approved by the ethical committee of IDIBELL.

The fibroblasts were grown in Dulbecco's modified Eagle's medium (Gibco, Thermo Fisher Scientific Inc., Rockford, IL, USA) containing 10% fetal bovine serum (Cultek, Ref. 91S1800; Madrid, Spain), 100 U/ml penicillin, and 100 μg/ml streptomycin (Pen Strep; Gibco, Ref. 15140-122) and maintained at 37°C in a humidified 95% air/5% $CO_2$ incubator. The compounds tested were added at 80–90% cell confluence. 15 μM oligomycin and 50 μM C26:0 were diluted in ethanol and added for 18 and 24 h, respectively. DMF (20 μM), dissolved in ethanol, and GSK-3β inhibitors (CHIR99021 at 3 μM and SB216763 at 10 μM), dissolved in dimethylsulfoxide (DMSO), were added 18 h after C26:0 treatment for 6 h. All experiments were performed with fibroblasts at passage 10–20.

## Nuclear-cytoplasmic fractionation in human fibroblasts

We performed subcellular fractionation to study NRF2 translocation to the nucleus, using a non-ionic detergent lysis method with slight modifications (Abmayr et al, 2006). Briefly, cells grown in 100 mm diameter dishes (Nunc Dish, Thermo Fisher Inc.) were washed in ice-cold phosphate buffer saline (PBS) pH 7.4 and collected by trypsinization (0.25% Trypsin-EDTA Solution A; Biological Industries USA, Cromwell, CT, USA) in a 15-ml falcon tube. After centrifugation and a new ice-cold PBS wash, the cell pellet was resuspended with 140 μl of lysis buffer (0.1% Nonidet P-40 in PBS, plus proteases (Complete Mini, Ref. 11836153001; Roche Diagnostics GmbH; Mannheim; Germany) and phosphatases (PhosSTOP, Ref. 04906845001; Roche Diagnostics GmbH) inhibitors) and scratched 20 times to lyse the cells. After 10-min incubation on ice, we centrifuged at 300 g during 5 min at 4°C in an Eppendorf® microcentrifuge. Supernatant was collected as cytoplasmic fraction. Then, we added 70 μl of RIPA buffer (50 mM Tris pH 8.0, 150 mM NaCl, 12 mM deoxycholic acid, and 1% Nonidet P-40; complemented with protease/phosphatase inhibitors) to the pellet to obtain the nuclear fraction. After homogenizing the pellet through a syringe with a 25G needle and 30 min of shaking at 4°C, we centrifuged for 10 min at 16,100 g at 4°C. Supernatant was collected as nuclear fraction. Then, we performed immunoblot procedures as described below. Lamin B1 and aldolase A were used as markers for nuclear and cytoplasmic fraction, respectively.

## Quantitative real-time PCR

RNA extraction and retrotranscription into cDNA, DNA extraction and quantitative RT–PCR analysis were performed as previously described (Morato et al, 2013). Total RNA was extracted from human fibroblasts and mouse tissues using RNeasy Kit (Qiagen, Hilden, Germany). Total DNA was extracted from mouse spinal cord using Gentra Puregene Tissue Kit (Qiagen, Hilden, Germany). The expression of the genes of interest was analyzed by Q–PCR using TaqMan® Gene Expression Assays (Thermo Fisher Scientific Inc.) and standardized TaqMan® probes (Appendix Table S2) on a Light-Cycler® 480 Real-Time PCR System (Roche Diagnostics GmbH). Relative quantification was carried out using the "Delta-Delta Ct" (ΔΔCt) method with *Rplp0* as endogenous control. To quantify mouse mitochondrial DNA (mtDNA) content, primers for mouse cytochrome b (*Cytb*) were designed (Custom TaqMan Gene Expression Assays; Thermo Fisher Scientific Inc.). The sequences for *Cytb* primers were as follows: ATGACCCCAATACGCAAAATTA (forward) and GGAGGACATAGCCTATGAAGG (reverse), and the FAM-labeled probe sequence was TTGCAACTATAGCAACAG. Quantification of mtDNA was referred to nuclear DNA (nDNA), determined by the amplification of the intron-less mouse nuclear gene *Cebpa* (Morato et al, 2013). Transcript quantification was performed in triplicate for each sample.

## Immunoblot

Human fibroblasts and mouse tissues were homogenized in RIPA buffer and then sonicated, centrifuged, and heated for 10 min at 70°C after adding 4X NuPAGE® LDS Sample Buffer (Invitrogen, Thermo Fisher Scientific Inc.). 20–50 μg of proteins was loaded onto 8% Novex NuPAGE® SDS–PAGE gel system (Invitrogen, Thermo Fisher Scientific Inc.) and run for 60–90 min at 120 V in NuPAGE® MOPS SDS Running Buffer (Invitrogen, Thermo Fisher Scientific Inc.) supplemented with 5 mM sodium bisulfite (Ref. 243973, Sigma-Aldrich). SeeBlue® Plus2 Pre-stained (Invitrogen, Thermo Fisher Scientific Inc.) was used as a ladder.

Regarding the different AKT or GSK-3β phosphorylations, we run the same quantity of samples (processed at the same time) in several gels in parallel, always performing Ponceau staining and Y-tubulin immunoblotting to confirm equal loading. Resolved proteins were transferred onto nitrocellulose membranes using iBlot® 2 Gel Transfer Device (Invitrogen, Thermo Fisher Scientific Inc.). After blocking in 5% bovine serum albumin (BSA, Sigma-Aldrich) in 0.05% TBS-Tween (TBS-T) for 1 h at room temperature, membranes were incubated with corresponding diluted primary antibodies (Appendix Table S1) in 5% BSA in 0.05% TBS-T overnight at 4°C. Following incubation with diluted secondary antibody (Appendix Table S1) in 0.05% TBS-T for 1 h at room temperature, proteins were detected with ECL Western blotting analysis system (GE Healthcare, Buckinghamshire, UK), followed by exposure to CL-XPosure Film (Thermo Fisher Scientific Inc.) as earlier described (Galino et al, 2011). It is worth noting that the correct band for NRF2 is at 100 kDa, not at the predicted 68 kDa (Lau et al, 2013). Immunoblots were quantified by densitometry using ImageJ v1.50i (U. S. National Institutes of Health, Bethesda, MD, USA).

## ATP

ATP levels were measured by a chemiluminescence system using ATPlite 1step (PerkinElmer, Inc., Waltham, MA, USA), as already described (Galino et al, 2011).

## Evaluation of oxidative lesions

AASA, CML, CEL, and MDAL concentrations in total proteins from spinal cord homogenates were measured by gas chromatography/

mass spectrometry (GC/MS), as reported (Fourcade *et al*, 2008). The amounts of products were expressed as the ratio of micromole of AASA, CML, CEL, or MDAL per mol of lysine.

### Measurement of very long-chain fatty acids

Content of very long-chain fatty acids in total lipids from spinal cord was analyzed as methyl ester derivative by gas chromatography, as described before (Morato *et al*, 2013). Briefly, separation was performed by a DBWAX capillary column (30 m × 0.25 mm × 0.20 μm) in a GC System 7890A with a Series Injector 7683B and a FID detector (Agilent Technologies). The injection port was maintained at 220°C, and the detector at 250°C; the temperature program was 5 min at 145°C, then 2°C/min to 240°C with a hold of 10 min, then 0.5°C/min to 250°C, and finally hold at 250°C for 5 min. Identification of fatty acid methyl esters was made by comparison with authentic standards (Sigma and Larodan Fine Chemicals). Results are represented as fold change to WT mice.

### Immunohistochemistry

Spinal cords were embedded in paraffin, and serial sections (4 μm thick) were cut in a transversal or longitudinal (1 cm long) plane after perfusion with 4% PFA. Immunohistochemistry (IHC) studies performed in WT, $Abcd1^{-}/Abcd2^{-/-}$ (DKO), and $Abcd1^{-}/Abcd2^{-/-}$ mice treated with DMF (DKO+DMF) were carried out using the avidin–biotin peroxidase method, as reported earlier (Launay *et al*, 2015).

After primary antibody incubation, the sections were incubated with the Labelled Streptavidin-Biotin2 System (LSAB2, Ref. K0675, Dako). Staining was visualized after incubation with 3,3′-diaminobenzidine (DAB) substrate chromogen (Ref. D5637, Sigma-Aldrich), which results in a brown-colored precipitate at the antigen site. After dehydrating the sections, slides were mounted with DPX (Ref. 06522, Sigma-Aldrich).

Images were acquired using Olympus BX51 microscope (20x/ N.A 0.50 Ph 1 UPlan FL N; Olympus Corporation, Tokyo, Japan) connected to an Olympus DP71 camera and Cell^B software (Olympus Corporation). The researcher was blinded to both genotype and treatment of the sample when analyzing the results. The number of $GFAP^{+}$ cells (astrocytes) and $Iba1^{+}$ cells (microglia) per mm² was determined in the spinal cord's ventral horn of WT, $Abcd1^{-}/Abcd2^{-/-}$, and DMF-treated $Abcd1^{-}/Abcd2^{-/-}$ mice ($n = 5$). The number of brown-colored cells was considered and counted with Cell Counter ImageJ plugin. Data are presented as an average of two 20× images per animal for each group.

### Analysis of locomotion

Locomotor deficits were assessed with the bar cross test and the treadmill test, as already described (Morato *et al*, 2013).

### Statistical analysis

Sample size was chosen according to previous experience in the laboratory, in similar experiments with long-term oral treatments that were performed with the same animal model. No animals were excluded from analysis. Animals and samples were allocated to

### The paper explained

**Problem**

X-linked adrenoleukodystrophy (X-ALD) is a rare disease that met the public eye in the early 90s thanks to the movie Lorenzo's Oil which has, despite recent advances in gene therapy, still no satisfactory treatment for most cases. The underlying genetic defect causes the malfunction of the fatty acid transporter ABCD1. Because early hallmarks of X-ALD are oxidative damage and bioenergetic impairment, here we evaluated the endogenous antioxidant response seeking for suitable drug targets.

**Results**

Using mouse models of X-ALD and cells obtained from the skin of X-ALD patients, we uncovered an impaired NRF2 antioxidant response caused by aberrant activity of GSK-3β, a kinase upstream in this cascade. We found that GSK-3β inhibitors reactivated the blunted NRF2 response in patients' fibroblasts. In the mouse models ($Abcd1^{-}$ and $Abcd1^{-}/Abcd2^{-/-}$ mice), oral administration of dimethyl fumarate (DMF/BG12/Tecfidera), an FDA-approved NRF2 activator, normalized molecular defects relative to ABCD1 deficiency such as (i) oxidative damage, (ii) mitochondrial depletion and bioenergetic failure, and (iii) neuroinflammation. Moreover, DMF halted axonal degeneration and locomotor disability in the mouse model.

**Impact**

This preclinical study identifies a druggable pathway underlying axonal degeneration in X-ALD and paves the way to use dimethyl fumarate in phase II clinical trials. The study highlights as well the potential of drugs targeting the GSK-3β/NRF2 axis for other axonal disorders with shared pathomechanisms.

different treatment groups by randomization. Researchers were blinded to the group assignment and to the animal number when performing the locomotor experiments until the data were processed for statistical analysis.

The values were expressed as the mean ± standard deviation (SD). The significant differences when comparing two groups were determined by a two-tailed unpaired Student's *t*-test (*$P < 0.05$, **$P < 0.01$, ***$P < 0.001$). When comparing more than two groups, significant differences were determined by one-way ANOVA followed by Tukey's/Dunnett's *post hoc* tests (*/#$P < 0.05$, **/##$P < 0.01$, ***/###$P < 0.001$) or Kruskal–Wallis non-parametric test followed by Dunn's *post hoc* test (\$$P < 0.05$, \$\$$P < 0.01$, \$\$\$$P < 0.001$), after verifying normality (Shapiro–Wilk test). Statistical analyses were performed using SPSS for Windows version 12.0.

**Expanded View** for this article is available online.

### Acknowledgements

We thank Laia Grau, Juanjo Martínez, and Cristina Guilera (Neurometabolic Diseases Laboratory, IDIBELL) for technical assistance. IDIBELL is part of the CERCA Institution (Centres de Recerca de Catalunya) of the Generalitat of Catalonia. This study was supported by grants from the Spanish Institute for Health Carlos III and "Fondo Europeo de Desarrollo Regional (FEDER), Union Europea, una manera de hacer Europa" [PFIS FI12/00457] to P.R-R., [FIS PI14/ 01115, FIS PI17/00134] to M.P.O., [FIS PI13/00584, FIS PI14/00328] to R.P., [FIS PI11/01043, FIS PI14/00410, FIS PI17/00916] to A.P., [Miguel Servet program CP11/00080, CPII16/00016, FIS PI15/00857] to S.F.; the European Commission [FP7-241622] to A.P., the European Leukodystrophy Association [ELA2012-

033C1] to A.P; the Autonomous Government of Catalonia [SGR 2017SGR696] to R.P. and [SGR 2014SGR1430; 2017SGR1206] to A.P.; and the Centre for Biomedical Research on Rare Diseases (CIBERER) to N.L. and M.R. Locomotor experiments were performed by the SEFALer unit F5 (CIBERER) led by A.P.

## Author contributions

MD, MFB, SF, and AP conceived the study. PR-R, NL, MR, NYC, and AN performed the experiments. PR-R, NL, MR, NYC, AN, MP-O, RP, IF, SF, and AP designed and/or interpreted aspects of the different experiments. AP led the project and acquired the main funding. PR-R, SF, and AP wrote the original draft. PR-R, SF, and AP reviewed and edited the manuscript. All the co-authors gave inputs on the manuscript.

## Conflict of interest

The authors declare that they have no conflict of interest.

## For more information

(i) X-ALD in OMIM: http://omim.org/entry/300100

(ii) X-ALD database: http://adrenoleukodystrophy.info

(iii) European Leukodystrophy Association: http://ela-asso.com/en/

(iv) Author's website: http://www.neurometabolic-lab.org/

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
