## [Review Process File · EMBO Molecular Medicine]

Aberrant regulation of the GSK-3 β /NRF2 axis unveils a novel therapy for adrenoleukodystrophy

Pablo Ranea-Robles, Nathalie Launay, Montserrat Ruiz, Noel Ylagan Calingasan, Magali Dumont, Alba Naudí, Manuel Portero-Otín, Reinald Pamplona, Isidre Ferrer, M. Flint Beal, Stéphane Fourcade and Aurora Pujol

Review timeline:

Submission date:	20 October 2017
Editorial Decision:	21 December 2017
Revision received:	22 March 2018
Editorial Decision:	16 May 2018
Revision received:	06 June 2018
Accepted:	12 June 2018

Editor: Céline Carret

Transaction Report:

1st Editorial Decision

21 December 2017

Thank you for the submission of your manuscript to EMBO Molecular Medicine. Please accept my apologies for getting back to you with such unusual delay. We have now heard back from the three referees whom we asked to evaluate your manuscript.

You will see from the set of comments pasted below that all referees found the data of interest and suitable for EMBO Molecular Medicine, in principle. Referees 2 and 3 only have minor comments but still relevant in terms of literature citation, M&M and results with more appropriate reporting. Referee 1 however, is a bit more critical and would like to see a more moderated discussion to reflect the limitations due to this compound, but also to potentials/hurdles for human translation (given the drug specificity for ALD or not, but importantly also the double KO mouse model used); this referee also would like to see whether the drug affects C26:0 and ideally speaking compare effects to other GSK inhibitors, which we agree would greatly add to the paper.

Upon our cross-commenting exercise, expressing moderation to reflect the study limitations is encouraged, as well as showing how DMF impacts C26:0 itself or whether the aggravation seen with C26:0 is ameliorated after DMF.

We would therefore welcome the submission of a revised version within three months for further consideration and would like to encourage you to address all the criticisms raised as suggested to improve conclusiveness and clarity. Please note that EMBO Molecular Medicine strongly supports a single round of revision and that, as acceptance or rejection of the manuscript will depend on another round of review, your responses should be as complete as possible.

I look forward to receiving your revised manuscript.

***** Reviewer's comments *****

Referee #1 (Comments on Novelty/Model System for Author):

The use of the double knockout mouse is a poor model system as no patients have ever been reported to harbor ABCD2 mutations.

Referee #1 (Remarks for Author):

This is a first paper reporting on an impaired NRF2 antioxidant response due to ABCD1 deficiency in adrenoleukodystrophy (ALD). The authors demonstrate that this is caused by aberrant activity in the GSK-3beta pathway and employ both *Abcd1* knockout and *Abcd1/Abcd2* double knockout mice to show this. Lastly they use dimethyl fumarate (DMF) as NRF2 activator to rescue pathology in the model system.

While the report is novel and carries potential clinical importance, there are gaps that the authors should address. The oxidative stress markers are shown to worsen with C26:0 (Fig 2), the biochemical hallmark of ABCD1 deficiency and presumed culprit of pathology. However, there are no data showing how DMF impacts C26:0 itself or whether the aggravation seen with C26:0 is ameliorated after DMF. Overall the use of the double knockout mouse is a poor model system as no patients have ever been reported to harbor ABCD2 mutations. This should be clearly stated as limitation in the discussion. If C26:0 levels are not impacted by DMF in the spinal cord, this should also be clearly stated.

Have the authors directly compared DMF to other GSK inhibitors? The authors may thereby demonstrate independent mechanisms pertinent to ABCD1 deficiency. When reading the discussion one cannot help wonder why DMF has not shown broad success across degenerative diseases in humans. It would be important to point to limitations of the compound when it come to translation to humans.

Minor comment: introduction - inflammatory demyelinating disease occurs not only in children but also in adults.

Referee #2 (Remarks for Author):

The manuscript by Ranea-Robles et al describes the identification of an impaired NRF2 antioxidant response caused by an aberrant activity in the GSK-3beta axis.

The authors find that DMF, an inhibitor of GSK-3b and an activator of NRF2 restores VLCFA-induced mitochondrial dysfunction and oxidative damage. Oral treatment of ALD mice with DMF corrects the phenotype, halts axonal degeneration and improves locomotor function.

This is a really well-written, pleasantly readable manuscript with very convincing and encouraging data. Therefore, it is a pity, and unnecessary, that the authors have a high tendency to basically only cite their own work when dealing with ALD literature. It appears that they have succeeded to cite almost every publication they ever published themselves, but they leave out key ALD publications.

For example: there is no good reference with respect to ALD biochemistry. There are no references with respect to the clinical aspect of ALD. There is no reference to the fact that heterozygous

women develop clinical symptoms resembling AMN. There are no references to the lack of a genotype-phenotype correlation.

And when there is a reference to ALD work it's often a wrong citation. For example: Shinnoh et al 1995 did not demonstrate that ALDP transports VLCFA into peroxisomes. This study demonstrated that the biochemical defect in ALD cells can be corrected by transfecting the normal ABCD1 cDNA. The correct references are van Roermund FASEB 2008 and Wiesinger JBC 2013.

Furthermore, the Pierpont 2017 paper is not a proper reference to bone marrow transplantation. This paper discusses the long-term neurocognitive outcomes of transplanted boys. Alternatives are Aubourg NEJM 1990, Miller Blood 2011 and others.

Finally, VLCFA do not only cause oxidative stress. The recent paper demonstrating the VLCFA induce ER stress and that there is a clear correlation between fatty acid length and ER stress should be mentioned too (van de Beek 2017).

Importantly, there is a discrepancy between the materials and methods (page 21) and the legends of Figures 3 and 4. In the M&M the authors state that the number of mice that are treated with DMF to characterize the effect on biochemical signs is n=12 per group. However, in the legend of the corresponding Figures 3 and 4 there are only 6 mice per group. The authors should explain what happened with the other 6 mice? It's hard to imagine that this is a typo.

Referee #3 (Comments on Novelty/Model System for Author):

It would be important to see the entire western blots; It is very difficult to reprobe the blots for the same AKT that only recognize different phosphorylation sites. However I do not know how well these antibodies work on mice and man, as they were used.

Referee #3 (Remarks for Author):

The manuscript entitled "Aberrant regulation of the GSK-3beta NRF2 axis unveils a novel therapy for adrenoleukodystrophy" by Ranea-Robles et al describe a clear effect of DMF on the mouse model for X-ALD as well as in human fibroblasts. Manifold different impairments known to be present in the spinal cord of the mouse model of X-ALD at the age of 12 months could be confirmed which is per se of greatest importance for the community. Moreover, all of which could be completely restored to wt levels using the oral treatment of DMF. These results are very impressive.

Minor comments:

Fig 1C and D as well as Fig 2D and 2E: Was this a reprobing of the very same blot for 7 or 6 times? Please indicate in Materials and Methods if and how it was ensured that it was completely striped in between the different phosphospecific AKA antibodies. Please show the entire blots for all samples in the supplementary data.

Fig 5: please perform also a quantification for Iba1 and GFAP as it was done for Synaptophysin and APP. The N should be given per e.g. square millimetre. In materials and methods it should be indicated how the quantification was performed.

1st Revision - authors' response

22 March 2018

Referee #1 (Comments on Novelty/Model System for Author):

Q: The use of the double knockout mouse is a poor model system as no patients have ever been reported to harbor ABCD2 mutations.

A: We thank the referee for his/her helpful comments and the constructive criticism of our manuscript. Regarding this comment, Referee#1 is right and there is no reported mutation of *ABCD2* in humans, however its overlapping role regarding import of C26:0 into peroxisomes has been demonstrated (Fourcade et al, 2009), and its potential role as genetic modifier of the metabolic impairment in XALD patients has been postulated (Muneer et al, 2014).

We thus think that the double knockout mouse *Abcd1^{-/-}/Abcd2^{-/-}* still represents a suitable preclinical

model to test treatments, and it is the best mouse model available. These mice present an AMN-like phenotype similar to the phenotype displayed by *Abcd1* mice, but with an earlier onset of axonal degeneration and a more severe locomotor disability at 12 months of age (Pujol et al, 2004), which allows to assess more consistently effects of drugs trials. It is well known that wild type mice older than 20 months may present aging-related pathology, what makes it very difficult to discriminate them from the 20 months *Abcd1* null, given their mild phenotype, which implies the use of very high numbers of animals for reliable locomotor task assessment. This is impractical due to economic and structural constraints of the vast majority of research institutions world-wide, including ours. For all these reasons, the double knockout has been used extensively for testing therapeutic approaches by our lab (Launay et al, 2015; Launay et al, 2017; Lopez-Erauskin et al, 2011; Morato et al, 2013; Morato et al, 2015) and others (Mastroeni et al, 2009). Some of the compounds used in these mice have made all the way to clinical trials (Lopez-Erauskin et al, 2011; Morato et al, 2013). We have included a paragraph in the Introduction section explaining this point (pages 5-6, lines 99-107: “The mouse model of X-ALD (*Abcd1* null), develops axonopathy and locomotor impairment very late in life, at 20 months of age, resembling adrenomyeloneuropathy, the most frequent X-ALD phenotype (Pujol et al, 2002). The closest homolog *Abcd2* exhibits overlapping metabolic functions (Fourcade et al, 2009) and has been postulated as modifier of the biochemical defect (Muneer et al, 2014).

Double mutant *Abcd1*^{-/-}/*Abcd2*^{-/-} mice develop a more severe, earlier onset axonopathy starting at 12 months of age, what makes them a more suitable model for therapeutic essays (Launay et al, 2015; Launay et al, 2017; Lopez-Erauskin et al, 2011; Mastroeni et al, 2009; Morato et al, 2013; Morato et al, 2015; Pujol et al, 2002)”

Referee #1 (Remarks for Author):

Q: This is a first paper reporting on an impaired NRF2 antioxidant response due to ABCD1 deficiency in adrenoleukodystrophy (ALD). The authors demonstrate that this is caused by aberrant activity in the GSK3beta pathway and employ both *Abcd1* knockout and *Abcd1/Abcd2* double knockout mice to show this. Lastly they use dimethyl fumarate (DMF) as NRF2 activator to rescue pathology in the model system.

While the report is novel and carries potential clinical importance, there are gaps that the authors should address. The oxidative stress markers are shown to worsen with C26:0 (Fig 2), the biochemical hallmark of ABCD1 deficiency and presumed culprit of pathology. However, there are no data showing how DMF impacts C26:0 itself or whether the aggravation seen with C26:0 is ameliorated after DMF. Overall the use of the double knockout mouse is a poor model system as no patients have ever been reported to harbor ABCD2 mutations. This should be clearly stated as limitation in the discussion. If C26:0 levels are not impacted by DMF in the spinal cord, this should also be clearly stated.

Have the authors directly compared DMF to other GSK inhibitors? The authors may thereby demonstrate independent mechanisms pertinent to ABCD1 deficiency.

A: We are very grateful to the referee for these suggestions that we agree will improve the final message of this manuscript. Regarding the question about DMF effect on the aggravation seen with C26:0 in the human fibroblasts, we have performed Q-PCR analyses to measure the expression of the three NRF2 target genes used to identify an NRF2-blunted response on X-ALD fibroblasts (*HMOX1*, *NQO1* and *GCLC*). To compare DMF with GSK-3β inhibitors, we evaluated simultaneously the effect of DMF or GSK-3β inhibitors on the aggravation seen with C26. The dose of DMF selected was 20 μM, similar to what have been published (Lastres-Becker et al, 2016). We increased the n with three new controls and three new X-ALD fibroblasts, and we made the corresponding change in materials and methods (page 22, line 474 and lines 484-486: DMF (20 μM), dissolved in ethanol, and GSK-3β inhibitors (CHIR99021 at 3μM and SB216763 at 10 μM), dissolved in dimethylsulfoxide (DMSO), were added 18 hours after C26:0 treatment for 6 hours). We observed a similar effect of DMF compared with GSK-3β inhibitors. DMF reactivated the transcriptional activation of *HMOX1*, *NQO1* and *GCLC*. Furthermore, DMF alone induced the expression of *HMOX1* and *NQO1* in control and *HMOX1* in X-ALD fibroblasts. Thus, these new data reinforce the rationale for DMF treatment *in vivo*, and we have subsequently included these data in the results section (page 10, line 198-203: “Before treating the animals, we tested DMF in control and X-ALD fibroblasts. We found that DMF reactivated the NRF2-blunted response upon VLCFA addition (Fig. EV2), similar to the GSK-3β inhibitors used (Fig 2F). Moreover, DMF alone induced

HMOX1 and *NQO1* expression in control fibroblasts, and also *HMOX1* expression in X-ALD fibroblasts (Fig. EV2). Thus, these new data reinforced the rationale for DMF treatment *in vivo*). We present this result (only DMF treatment) in a new **Expanded View Figure 2**.

After increasing the number of fibroblasts used, the results with GSK-3 β inhibitors are more robust now and led to a new **Fig. 2F**, replacing the former one, and presented below. Like we report in the manuscript, GSK-3 β inhibitors (C99021 and SB 216763) rescued the blunted antioxidant response against C26:0 in X-ALD fibroblasts. Even though it would also be of interest to compare the effects of DMF and GSK-3 β inhibitors *in vivo*, we have decided not to perform these experiments given that generating the mice at the proper age and performing the experiments would take more than year, thus largely surpassing the allocated time for the manuscript's revision.

Figure EV2. DMF effect on NRF2 response in X-ALD fibroblasts

Expression of NRF2 target genes was measured in CTL (n=3) and X-ALD (n=3) after C26:0 (50 μ M, 24h) and/or DMF (20 μ M, 6h). Gene expression normalized relative to *RPLP0*. Quantification depicted as fold change to vehicle-treated (Veh) fibroblasts.

Data information: Data are presented as mean \pm SD. *P<0.05, **P<0.01, ***P<0.001 (one-way ANOVA followed by Tukey's post-hoc test)

Fig. 2F

NRF2-dependent antioxidant gene expression (*HMOX1*, *NQO1* and *GCLC*) after GSK-3 β inhibition in VLCFA-treated CTL (n=8 per condition) and X-ALD (n=8 per condition) fibroblasts. Gene expression normalized relative to *RPLP0*. Quantification depicted as fold change to vehicle-treated (Veh) fibroblasts. Data information: Data are presented as mean \pm SD. *P<0.05, **P<0.01, ***P<0.001 (one-way ANOVA followed by Tukey's post-hoc test). \$P<0.05, \$\$P<0.01 (non-parametric Kruskal-Wallis' test followed by Dunn's post-hoc test).

We have also measured very long-chain fatty acids, C24:0 and C26:0, levels in the spinal cord of WT, *Abcd1*^{-/-}, and *Abcd1* mice after DMF treatment (n=5 per condition), by gas chromatography (Morato et al, 2013). DMF treatment did not modify the levels of VLCFA in the spinal cord of *Abcd1* mice, reflected by C24:0/C22:0 and C26:0/C22:0 ratios. The results are presented below, and included in a new **Expanded View Figure 3**, as well as in results section (page 11, lines 224-225: "These effects seem to be independent of VLCFA levels, since DMF treatment did not alter C24:0 or C26:0 levels in the spinal cord of 12-month-old *Abcd1* mice (Fig EV3)"). The corresponding

method has been added to Materials and Methods section (pages 25-26, lines 559-569):
 “**Measurement of very long-chain fatty acids**

Content of very long-chain fatty acids in total lipids from spinal cord was analysed as methyl ester derivative by gas chromatography, as described before (Morato et al, 2013). Briefly, separation was performed by a DBWAX capillary column (30 m • 0.25 mm • 0.20 µm) in a GC System 7890A with a Series Injector 7683B and a FID detector (Agilent Technologies). The injection port was maintained at 220°C, and the detector at 250°C; the temperature program was 5 min at 145°C, then 2°C/min to 240°C with a hold of 10 min, then 0.5°C/min to 250°C, and finally hold at 250°C for 5 min. Identification of fatty acid methyl esters was made by comparison with authentic standards (Sigma and Larodan Fine Chemicals). Results were represented as fold change to WT mice.”)

Figure EV3. VLCFA levels after DMF treatment in the spinal cord of X-ALD mouse model

Saturated fatty acids 22:0, 24:0 and 26:0 level in the spinal cord of 12-month-old WT, *Abcd1*^{-/-} and *Abcd1*^{-/-} + DMF mice (n=5). Fatty acid quantification was done by gas-liquid chromatography (see materials and methods). VLCFA 24:0 and 26:0 levels are normalized to the long-chain fatty acid 22:0 in the 26:0/22:0 and the 24:0/22:0 ratios. Quantification is depicted as fold change to WT mice. Data information: Data are presented as mean ± SD. *P<0.05, **P<0.01, ***P<0.001 (one-way ANOVA followed by Tukey’s post hoc test)

Q: When reading the discussion one cannot help wonder why DMF has not shown broad success across degenerative diseases in humans. It would be important to point to limitations of the compound when it come to translation to humans.

A: DMF has shown an acceptable safety profile in the clinical trials for relapsing-remitting multiple sclerosis patients (Fox et al, 2012; Gold et al, 2012), but we agree with Referee #1 about the importance of commenting the limitations of the drug. Thus, we appreciate the suggestion and have included a paragraph in the discussion regarding side effects of DMF, but still we consider that the small number of cases (5 cases out of >230,000 treated MS patients) of progressive multifocal leukoencephalopathy (PML) after DMF treatment (Gieselbach et al, 2017) deserved a commentary in this section (pages 18-19, lines 395-407: “In view of these data, DMF exhibits a great potential to treat X-ALD and other neurodegenerative diseases with an overall good safety profile (Fox et al, 2012; Gold et al, 2012). However, some precautions need to be observed with this drug, as side effects need to be monitored and evaluated carefully. The most important is lymphocytopenia (Fox et al, 2016). Of >230,000 patients treated with DMF in the period of 3 years following commercial availability, five cases of progressive multifocal leukoencephalopathy (PML) have been reported, in the setting of moderate to severe prolonged lymphocytopenia (Pardo & Jones, 2017). As a consequence, FDA and EMA (EMA/627077/2015) recently issued updated safety recommendations to minimize PML risk, which include regular blood counts. Recent reports indicates that a reduction in T cells rather than a general reduction in lymphocyte count may be associated to these cases (Gieselbach et al, 2017), which still represent a very low percentage of all the patients treated with fumaric acid esters”)

Q: Minor comment: introduction - inflammatory demyelinating disease occurs not only in children but also in adults.

A: We appreciate this comment from Referee #1 and we realized that the sentence in introduction referring to inflammatory demyelination may lead to confusion. We have changed this sentence to specify that inflammatory demyelination can occur also in adolescents and adults (page 5, lines 86-89: “First, cerebral adrenoleukodystrophy is present mostly in boys between 5-10 years (35-40% of the cases) but also in adolescents and adult men, who present a strong inflammatory demyelinating reaction in central nervous system white matter.”)

Referee #2 (Remarks for Author):

Q: The manuscript by Ranea-Robles et al describes the identification of an impaired NRF2 antioxidant response caused by an aberrant activity in the GSK-3beta axis.

The authors find that DMF, an inhibitor of GSK-3b and an activator of NRF2 restores VLCFA-induced mitochondrial dysfunction and oxidative damage. Oral treatment of ALD mice with DMF corrects the phenotype, halts axonal degeneration and improves locomotor function.

This is a really well-written, pleasantly readable manuscript with very convincing and encouraging data. Therefore, it is a pity, and unnecessary, that the authors have a high tendency to basically only cite their own work when dealing with ALD literature. It appears that they have succeeded to cite almost every publication they ever published themselves, but they leave out key ALD publications.

For example: there is no good reference with respect to ALD biochemistry. There are no references with respect to the clinical aspect of ALD. There is no reference to the fact that heterozygous women develop clinical symptoms resembling AMN. There are no references to the lack of a genotype-phenotype correlation.

And when there is a reference to ALD work it's often a wrong citation. For example: Shinnoh et al 1995 did not demonstrate that ALDP transports VLCFA into peroxisomes. This study demonstrated that the biochemical defect in ALD cells can be corrected by transfecting the normal ABCD1 cDNA. The correct references are van Roermund FASEB 2008 and Wiesinger JBC 2013. Furthermore, the Pierpont 2017 paper is not a proper reference to bone marrow transplantation. This paper discusses the long-term neurocognitive outcomes of transplanted boys. Alternatives are Aubourg NEJM 1990, Miller Blood 2011 and others. Finally, VLCFA do not only cause oxidative stress. The recent paper demonstrating the VLCFA induce ER stress and that there is a clear correlation between fatty acid length and ER stress should be mentioned too (van de Beek 2017).

A: We agree with the referee #2 and we apologize for missing the mentioned references. We have added or changed suggested citations into the text (page 5). We have also included the ER stress study citation in the Results section (page 8, line 158).

Q: Importantly, there is a discrepancy between the materials and methods (page 21) and the legends of Figures 3 and 4. In the M&M the authors state that the number of mice that are treated with DMF to characterize the effect on biochemical signs is n=12 per group. However, in the legend of the corresponding Figures 3 and 4 there are only 6 mice per group. The authors should explain what happened with the other 6 mice? It's hard to imagine that this is a typo.

A: There is no typo in figure legends. We explain to Referee #2 how we used the mice: 12 mice were treated with DMF and all of them have been used, but we did not use all the animals in each experiment, due to sample exhaustion. The number of animals used per experiment has been specified in the figure legends.

Referee #3 (Comments on Novelty/Model System for Author):

Q: It would be important to see the entire western blots; It is very difficult to reprobe the blots for the same AKT that only recognize different phosphorylation sites. However I do not know how well these antibodies work on mice and man, as they were used.

A: We have added the entire immunoblots for every figure as Source Data. Regarding the incubation of the same membrane with different antibodies, we did not strip and reprobe the membranes. We

cut the membranes in several pieces and used different antibodies (combinations that differ enough in molecular weight to be used within the same membrane). Regarding the different AKT phosphorylations, we run the same quantity of samples (processed at the same time) in several gels in parallel, always performing Ponceau staining and γ -tubulin immunoblotting to confirm equal loading. To avoid confusion, we have added this information in a sentence in materials and methods (page 24, lines 535-537: “Regarding the different AKT or GSK-3 β phosphorylations, we run the same quantity of samples (processed at the same time) in several gels in parallel, always performing Ponceau staining and γ -tubulin immunoblotting to confirm equal loading.”) Antibodies for AKT and GSK-3 β immunoblotting were from Cell Signaling, as stated in materials and methods section. These antibodies have been cited hundreds of times in both mice and humans.

Referee #3 (Remarks for Author):

The manuscript entitled "Aberrant regulation of the GSK-3 β NRF2 axis unveils a novel therapy for adrenoleukodystrophy" by Ranea-Robles et al describe a clear effect of DMF on the mouse model for XALD as well as in human fibroblasts. Manifold different impairments known to be present in the spinal cord of the mouse model of X-ALD at the age of 12 months could be confirmed which is per se of greatest importance for the community. Moreover, all of which could be completely restored to wt levels using the oral treatment of DMF. These results are very impressive.

Minor comments:

Q: Fig 1C and D as well as Fig 2D and 2E: Was this a reprobing of the very same blot for 7 or 6 times? Please indicate in Materials and Methods if and how it was ensured that it was completely striped in between the different phosphospecific AKA antibodies. Please show the entire blots for all samples in the supplementary data.

A: We thank the referee for his appreciation of our work and this helpful remarks. This comment has been addressed just before.

Q: Fig 5: please perform also a quantification for Iba1 and GFAP as it was done for Synaptophysin and APP. The N should be given per e.g. square millimetre. In materials and methods it should be indicated how the quantification was performed.

A: We have quantified Iba1 and GFAP positive cells per squared millimetre, as suggested by Referee #3. The result has been added as two new graphs to a new Figure 5Y, and commented on results section (page 13, lines 276-277: “as it normalized the density of astrocytes and microglial cells in DKO mice (Fig 5A-F and Y)”). We are grateful to the referee for this suggestion, as the result is more robust with this quantification. Microgliosis and astrocytosis are reversed by DMF treatment. We have also added how this quantification was performed to materials and method section (pages 26-27, Lines 583-588: “The number of GFAP⁺ cells (astrocytes) and Iba1⁺ positive cells (microglia) per mm² was determined in the spinal cord’s ventral horn of WT, *Abcd1*^{+/+}/*Abcd2*^{-/-} and DMF-treated *Abcd1*^{-/-}/*Abcd2*^{-/-} mice (n=5). The number of brown coloured-cells was considered and counted with Cell Counter Image J plugin. Data are presented as an average of two 20x images per animal for each group.”)

New graphs of figure 5Y: Quantification of GFAP⁺ and Iba1⁺ cells per mm² in spinal cord immunohistological sections of WT (n=5), DKO (n=5) and DKO + DMF (n=5) mice.

Data information: Data are presented as mean \pm SD. **P<0.01, ***P<0.001 (one-way ANOVA followed by Tukey’s post-hoc test).

References

- Fourcade S, Ruiz M, Camps C, Schluter A, Houten SM, Mooyer PA, Pampols T, Dacremont G, Wanders RJ, Giros M et al (2009) A key role for the peroxisomal ABCD2 transporter in fatty acid homeostasis. *Am J Physiol Endocrinol Metab* 296: E211-221
- Fox RJ, Chan A, Gold R, Phillips JT, Selmaj K, Chang I, Novas M, Rana J, Marantz JL (2016) Characterizing absolute lymphocyte count profiles in dimethyl fumarate-treated patients with MS: Patient management considerations. *Neurol Clin Pract* 6: 220-229
- Fox RJ, Miller DH, Phillips JT, Hutchinson M, Havrdova E, Kita M, Yang M, Raghupathi K, Novas M, Sweetser MT et al (2012) Placebo-controlled phase 3 study of oral BG-12 or glatiramer in multiple sclerosis. *N Engl J Med* 367: 1087-1097
- Gieselbach RJ, Muller-Hansma AH, Wijburg MT, de Bruin-Weller MS, van Oosten BW, Nieuwkamp DJ, Coenjaerts FE, Wattjes MP, Murk JL (2017) Progressive multifocal leukoencephalopathy in patients treated with fumaric acid esters: a review of 19 cases. *J Neurol* 264: 1155-1164
- Gold R, Kappos L, Arnold DL, Bar-Or A, Giovannoni G, Selmaj K, Tornatore C, Sweetser MT, Yang M, Sheikh SI et al (2012) Placebo-controlled phase 3 study of oral BG-12 for relapsing multiple sclerosis. *N Engl J Med* 367: 1098-1107
- Lastres-Becker I, Garcia-Yague AJ, Scannevin RH, Casarejos MJ, Kugler S, Rabano A, Cuadrado A (2016) Repurposing the NRF2 Activator Dimethyl Fumarate as Therapy Against Synucleinopathy in Parkinson's Disease. *Antioxid Redox Signal* 25: 61-77
- Launay N, Aguado C, Fourcade S, Ruiz M, Grau L, Riera J, Guilera C, Giros M, Ferrer I, Knecht E et al (2015) Autophagy induction halts axonal degeneration in a mouse model of X-adrenoleukodystrophy. *Acta Neuropathol* 129: 399-415
- Launay N, Ruiz M, Grau L, Ortega FJ, Ilieva EV, Martinez JJ, Galea E, Ferrer I, Knecht E, Pujol A et al (2017) Tauroursodeoxycholic bile acid arrests axonal degeneration by inhibiting the unfolded protein response in X-linked adrenoleukodystrophy. *Acta Neuropathol* 133: 283-301
- Lopez-Erauskin J, Fourcade S, Galino J, Ruiz M, Schluter A, Naudi A, Jove M, Portero-Otin M, Pamplona R, Ferrer I et al (2011) Antioxidants halt axonal degeneration in a mouse model of X-adrenoleukodystrophy. *Ann Neurol* 70: 84-92
- Mastroeni R, Bensadoun JC, Charvin D, Aebischer P, Pujol A, Raoul C (2009) Insulin-like growth factor-1 and neurotrophin-3 gene therapy prevents motor decline in an X-linked adrenoleukodystrophy mouse model. *Ann Neurol* 66: 117-122
- Morato L, Galino J, Ruiz M, Calingasan NY, Starkov AA, Dumont M, Naudi A, Martinez JJ, Aubourg P, Portero-Otin M et al (2013) Pioglitazone halts axonal degeneration in a mouse model of X-linked adrenoleukodystrophy. *Brain* 136: 2432-2443
- Morato L, Ruiz M, Boada J, Calingasan NY, Galino J, Guilera C, Jove M, Naudi A, Ferrer I, Pamplona R et al (2015) Activation of sirtuin 1 as therapy for the peroxisomal disease adrenoleukodystrophy. *Cell Death Differ* 22: 1742-1753
- Muneer Z, Wiesinger C, Voigtlander T, Werner HB, Berger J, Forss-Petter S (2014) Abcd2 is a strong modifier of the metabolic impairments in peritoneal macrophages of ABCD1-deficient mice. *PLoS One* 9: e108655
- Pardo G, Jones DE (2017) The sequence of disease-modifying therapies in relapsing multiple sclerosis: safety and immunologic considerations. *J Neurol* 264: 2351-2374
- Pujol A, Ferrer I, Camps C, Metzger E, Hindelang C, Callizot N, Ruiz M, Pampols T, Giros M, Mandel JL (2004) Functional overlap between ABCD1 (ALD) and ABCD2 (ALDR) transporters: a

therapeutic target for X-adrenoleukodystrophy. HumMolGenet 13: 2997-3006

2nd Editorial Decision

16 May 2018

Thank you for the submission of your revised manuscript to EMBO Molecular Medicine. We have now received the enclosed reports from the referee who was asked to re-assess it. As you will see the reviewer is supportive and I am pleased to inform you that we will be able to accept your manuscript pending the following final amendments:

1) Please address the last comment of referee 3. We would encourage you to complement the Source Data filer that you already have provided with the controls requested.

Please submit your revised manuscript within two weeks. I look forward to seeing a revised form of your manuscript as soon as possible.

***** Reviewer's comments *****

Referee #3 (Remarks for Author):

The authors have well coped to the reviewers suggestions and the manuscript has additionally improved. There is only one last point from my side with regard of Fig 1C. I do understand from the authors comment that this figure represents six individual blots equally loaded and processed in parallel. Thus, the addition of the control for equally lodging and transfer should be added for each individual blot. If the space in the figure 1 does not allow the adding of five additional panels, the lower TUB loading control, which only represent one of the 6 different blots, should be removed (to avoid any possible confuse the reader) and the full figure 1C with all controls can be shown in the Expanded View of the figure. In any case, variant 1 (show all controls within the original figure) would be much preferred.

2nd Revision - authors' response

06 June 2018

***** Reviewer's comments *****

Referee #3 (Remarks for Author):

The authors have well coped to the reviewers suggestions and the manuscript has additionally improved. There is only one last point from my side with regard of Fig 1C. I do understand from the authors comment that this figure represents six individual blots equally loaded and processed in parallel. Thus, the addition of the control for equally lodging and transfer should be added for each individual blot. If the space in the figure 1 does not allow the adding of five additional panels, the lower TUB loading control, which only represent one of the 6 different blots, should be removed (to avoid any possible confuse the reader) and the full figure 1C with all controls can be shown in the Expanded View of the figure. In any case, variant 1 (show all controls within the original figure) would be much preferred.

Thanks for the comment. The figure 1C represents 3 individual membranes instead of 6, equally loaded and processed in parallel, incubated with two antibodies each. Each blot or membrane has been incubated with total AKT, total GSK-3 β or their phosphorylated forms, always with γ -tubulin as loading control either in parallel or sequentially depending on the precise antibody. We have included the 2 additional γ -tubulin blots in the Source Data Fig 1C, as suggested by the referee #3 and the journal's editor.

Corresponding Author Name: Aurora Pujol
 Journal Submitted to: EMBO Molecular Medicine
 Manuscript Number: EMM-2017-08604